# Estimating vadose zone water fluxes from soil water monitoring data: a comprehensive field study in Austria

Marleen Schübl[1], Giuseppe Brunetti[1], Gabriele Fuchs[2], and Christine Stumpp[1]

[1]University of Natural Resources and Life Sciences, Vienna, Department of Water, Atmosphere and Environment, Institute of Soil Physics and Rural Water Management, Muthgasse 18, 1190 Vienna, Austria
[2]Federal Ministry of Agriculture, Regions and Tourism (BMLRT), Division I/3: Department of Water Resources, Marxergasse 2, 1030 Vienna, Austria

**Correspondence:** Marleen Schübl (marleen.schuebl@boku.ac.at )

**Abstract.** Groundwater recharge is a key component of the hydrological cycle, yet its direct measurement is complex and often difficult to achieve. An alternative is its inverse estimation through a combination of numerical models and transient observations from distributed soil water monitoring stations. However, an often neglected aspect of this approach is the effect of model predictive uncertainty on simulated water fluxes. In this study, we made use of long-term soil water content measurements at 14 locations from the Austrian soil water monitoring program to quantify and compare local, potential groundwater recharge rates and their temporal variability. Observations were coupled with a Bayesian probabilistic framework to calibrate the model HYDRUS-1D and assess the effect of model predictive uncertainty on long-term simulated recharge fluxes. Estimated annual potential recharge rates ranged from 44 mm a$^{-1}$ to 1319 mm a$^{-1}$ with a relative uncertainty (95% interquantile range/median) in the estimation between 1-39%. Recharge rates decreased longitudinally, with high rates and lower seasonality at western sites and low rates with high seasonality and extended periods without recharge at the southeastern and eastern sites of Austria. Higher recharge rates and lower actual evapotranspiration were related to sandy soils; however, climatic factors had a stronger influence on estimated potential groundwater recharge than soil properties, underscoring the vulnerability of groundwater recharge to the effects of climate change.

## 1 Introduction

Groundwater is the largest reservoir of liquid freshwater on earth and one of the most important sources of drinking and irrigation water. Under changing climatic conditions with extremes occurring more frequently and intensely, the strategic importance of groundwater for global water and food security is expected to further increase (Taylor et al., 2013). In some countries, such as Austria, groundwater including spring water is the most important water resource, making up 100% of the water supply (Vogel, 2001). The major limitation for sustainable groundwater use is recharge, which represents the maximum amount of water that may be withdrawn from an aquifer without depleting it. This makes it a crucial variable for groundwater resource management (Moeck et al., 2020; Taylor et al., 2013). A large portion of groundwater recharge comes from water infiltrating soil and flowing through the vadose zone towards the water table (Döll and Fiedler, 2008; Nolan et al., 2007). Infiltration capacity, root water uptake and evaporation from the upper soil layers determine the net amount of water which is

transported into the deeper vadose zone, following the gradient in matric potential and gravity (Vereecken et al., 2008). Water flow through the vadose zone is supposed to have a major influence on the process of groundwater recharge even at karst mountain sites (Berthelin et al., 2020; Hartmann et al., 2014; Kaminsky et al., 2021; Neukum et al., 2008).

The quantification of recharge is complicated by temporal and spatial variability and by the fact that direct measurements are difficult (Moeck et al., 2020, 2018; Nolan et al., 2007; Scanlon et al., 2002). Lysimeters are the only means to obtaining local measurements of seepage flow, which can be considered a good indicator of groundwater recharge (Moeck et al., 2020, 2018; Seneviratne et al., 2012; von Freyberg et al., 2015). However, their appropriate set up is difficult without introducing a bias in the hydrological processes (Barkle et al., 2011; Groh et al., 2016; Pütz et al., 2018; Stumpp et al., 2012). Furthermore, the operation and maintenance of lysimeters is expensive, which is why long-term lysimeter measurements are scarce (Nolz et al., 2016; von Freyberg et al., 2015). Among the most widely used alternatives for recharge estimation are methods based on artificial and environmental tracer experiments (e.g., Boumaiza et al., 2020; Chesnaux and Stumpp, 2018; Koeniger et al., 2016) and groundwater table fluctuations (Moeck et al., 2020; Collenteur et al., 2021). Common water table fluctuation methods, however, face some limitations in reflecting and predicting the actual recharge process (Collenteur et al., 2021; Healy and Cook, 2002).

Moeck et al. (2020) collected and investigated a global scale data set of natural groundwater recharge rates where, however, recharge rates from high altitudes were underrepresented. For mountain sites in particular, there is a lack of reported groundwater recharge rates (Bresciani et al., 2018; Moeck et al., 2020). A limited number of studies report local or regional recharge rates based on different modeling approaches using field measurements, such as groundwater levels and river discharge, or available information on vegetation and subsurface, and assess controlling factors on groundwater recharge (e.g., Barron et al., 2012; Collenteur et al., 2021; Hartmann et al., 2017; Keese et al., 2005; Neukum and Azzam, 2012).

An alternative is the inverse estimation of recharge fluxes through the unsaturated zone by calibrating vadose zone hydrological models against transient observations (e.g., soil water content and pressure head). Over the last decades, numerical modeling of soil water fluxes has been applied and improved, resulting in today's state of the art soil models with an implementation of the Richards Equation for simulating the transport of water through the soil, considering heat and energy balances and accounting for relevant processes such as plant water uptake and snow hydrology (Šimůnek et al., 2016, 2003; Vereecken et al., 2016).

The core of this modeling approach is generally the inverse estimation of hydraulically relevant parameters, such as Soil Hydraulic Parameters (SHPs) (e.g., Van Genuchten, 1980). The use of field measurements guarantees a higher generalizability of estimated parameters compared to small scale measurements of soil samples in the laboratory (Dyck and Kachanoski, 2010; Groh et al., 2018; Stumpp et al., 2012; Vereecken et al., 2008; Vrugt et al., 2008; Wöhling et al., 2008). Several studies have evaluated the use of vadose zone measurements for the inverse estimation of effective SHPs and the reliable prediction of recharge fluxes (Durner et al., 2008; Groh et al., 2018; Schelle et al., 2012). However, inverse parameter estimation is often treated as an optimization problem aiming at a unique solution, which neglects the uncertainty which is fundamentally associated with parameter identification. Uncertainties originate from different error sources including model input and forcing data, the initial and boundary conditions, the model structure, heterogeneity and scale effects (Beven, 2006; Vereecken et al.,

2016). Further, the quality and scope of calibration data affects the uncertainty in parameter estimation. It is important not to neglect uncertainties related to the model calibration as they can lead to uncertain or even failing predictions (Finsterle, 2015; Vrugt and Sadegh, 2013). The emergence of computationally efficient algorithms makes it possible to deal with uncertainties in a statistically rigorous way based on the Bayesian approach to statistics (e.g., Brunetti et al., 2019; Scharnagl et al., 2011; Wöhling et al., 2008). This approach relies on the idea of integrating a priori knowledge of the system in the statistical inference, to combine it with observed data in order to derive the posterior probability distribution of parameter values, which can be used to quantify model uncertainty. Posterior parameter distributions also reflect the non-uniqueness and equifinality of parameter values.

In combination with a soil hydraulic model, an efficient algorithm is needed to compute posterior distributions with an iterative Monte Carlo approach and to allow for a clear convergence in a reasonable amount of time. Skilling (2006) introduced Nested Sampling as an efficient Monte Carlo method to estimate the integral of the Bayesian evidence, the denominator of Bayes theorem, and obtain posterior distributions as a side product. Its efficiency has been further increased with ellipsoidal Nested Sampling (Mukherjee et al., 2006). Finally, ellipsoidal rejection sampling, as proposed by Feroz et al. (2009) with the MULTINEST algorithm, is able to account efficiently for multimodal posterior distributions. A Bayesian statistical framework using a Nested Sampling approach in combination with a physically based soil water model and soil water monitoring measurements thus provides a powerful tool for a comprehensive characterization of the vadose zone at individual sites and the estimation of local water balances, including an assessment of the model uncertainties.

In this study, we made use of long-term volumetric soil water content measurements at 14 different locations from the Austria wide soil water monitoring program and integrated them in a Bayesian probabilistic framework with the MULTINEST algorithm to calibrate the hydrological model HYDRUS-1D at each location. We used this approach to account for the uncertainties inherently associated with the inverse parameter estimation, and we simultaneously assessed and propagated the model predictive uncertainty in simulated local potential groundwater recharge rates. All sites were modeled with the same approach on a similar data basis supporting comparability of the results. Site properties included a variety of soils and climatic conditions which allowed to investigate factors which influence the long-term soil water balances and temporal variability of potential groundwater recharge.

## 2 Material and methods

### 2.1 Austrian soil water monitoring program

The locations of 14 Austrian soil water monitoring sites are shown in Fig. 1(a). Figure 1(b) gives an overview over soil types according to the digital soil map of Austria (BFW, 2016). Figures 1(c) and 1(d) show long-term annual areal precipitation and actual evapotranspiration estimates (modified from Kling et al. (2007b) and Kling et al. (2007a), respectively). According to texture information (ÖNORM L 1050), the soil types at the measurement sites vary between sand and silt loam/loamy silt (11 – 88% sand, 12 – 75 % silt, and 0 – 32% clay). Details on altitude, geo-coordinates, soil textures, and measurement depths are given in the Appendix (Table A1). Zettersfeld, Gschlössboden and Sillianberger Alm are on the sub-alpine level in

the southwest of Austria, characterized by high contents in organic matter, coarse soil texture and/or high skeleton fraction; Leutasch, Achenkirch, Gumpenstein and Aichfeld-Murboden are at the montane level from western to central Austria with soil textures ranging between sand and loam; Pettenbach, Elsbethen and Lauterach are located at the foothill zone in western to central Austria with soil textures ranging from loam to loamy silt; Kalsdorf, Schalladorf, Lobau and Frauenkirchen are situated in the southern and eastern lowlands with sandy to loamy soil textures. Locations included in this study are horizontally even at the plot scale, and usually consisting of uncultivated grassland. In contrast, cultivation of alternating crops was carried out at the location Pettenbach, where details on the crop cover for calibration and validation periods were obtained from technical reports provided by the Upper Austrian Government (Land OÖ, 2013, 2014).

Long-term field measurements of volumetric soil water content, measured with Time Domain or Frequency Domain Reflectometry (TDR/FDR) over several years, partly since 1996, are carried out within the Austrian Soil Water Monitoring Program of the Federal Ministry of Agriculture, Regions and Tourism (BMLRT). Under this program, continuous measurements are conducted at various depth levels of soil profiles with the aim of providing standardized and quality assured measurement data. For inverse parameter estimation in this study, we selected calibration periods of around six months with sufficiently complete and plausible soil water content measurement series (Fig. A2 in the Appendix) and aggregated the data to a daily resolution. We used a model spin-up period of two months to relax the effect of initial conditions on the estimation procedure. The length of calibration periods was chosen to be similar for all sites, long enough to be informative for a range of soil water conditions. We excluded the winter season requiring the simulation of snow accumulation and melt processes as it increases the computational cost and numerical sensitivity of the simulations and introduces additional complexity and potential biases in the calibration. The use of spring-summer months, which have an alternation of wet-dry periods, is expected to increase the informativeness of soil water measurements. The monitoring program also offers composite matric potential measurements from tensiometers and gypsum blocks. The discontinuity of the data complicates the modeling and analysis, which is why they have not been used in this study. Validation periods were chosen to provide one year or more of continuous, plausible data. Snow hydrology was simulated for the model validation, as described in Sect. 2.2.1. Details on calibration and validation periods are summarized in Table A2. Several locations were equipped with lysimeters: At Leutasch and Pettenbach, in situ soil water content measurements were directly obtained from lysimeter set ups; in Gumpenstein, soil water content measurements were obtained from a soil profile next to a lysimeter cluster which provided long-term seepage measurements. Lysimeter measurements from Leutasch and Gumpenstein were used for additional validation of recharge rates.

## 2.2 Modeling theory

### 2.2.1 Water flow and root water uptake

The mechanistic model HYDRUS-1D (Šimůnek et al., 2016) was used to simulate water flow in the vadose zone profiles. HYDRUS-1D is a finite element model that numerically solves the one-dimensional Richards equation [Eq. (1)]

$$\frac{\partial \theta}{\partial t} = \frac{\partial}{\partial z} \left[ K(h) \left( \frac{\partial h}{\partial z} + 1 \right) \right] - S(h) \tag{1}$$

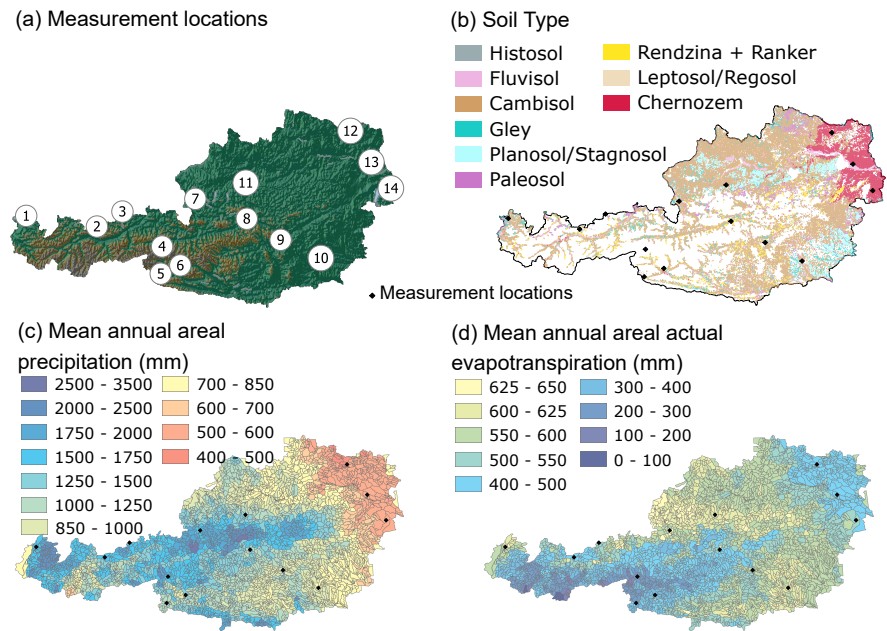

**Figure 1.** (a) Locations of 14 monitoring sites in Austria (1) Lauterach, (2) Leutasch, (3) Achenkirch, (4) Gschlössboden, (5) Sillianberger Alm, (6) Zettersfeld, (7) Elsbethen, (8) Gumpenstein, (9) Aichfeld-Murboden, (10) Kalsdorf, (11) Pettenbach, (12) Schalladorf, (13) Lobau, (14) Frauenkirchen; (b) Soil map data basis: Digital soil map of Austria, 1km raster, Federal Forest Research Center (BFW, 2016); (c) Hydrological Atlas of Austria (HAO) mean areal annual precipitation (Kling et al., 2007b); (d) HAO mean areal annual actual evapotranspiration (Kling et al., 2007a); Maps from the HAO where compiled using QGIS (QGIS Development Team, 2022).

125     where $\theta[L^3 L^{-3}]$ is the volumetric water content, $t[T]$ is the time variable, $z[L]$ is a vertical coordinate, $K(h)[LT^{-1}]$ is the unsaturated hydraulic conductivity function and $h[L]$ is the pressure head. $S[T^{-1}]$ is a sink term accounting for water uptake by plant roots. The unimodal Van Genuchten-Mualem (VGM) model described the soil hydraulic properties, namely the soil water retention curve [Eq. (2)], and the unsaturated hydraulic conductivity [Eq. (3)]:

$$\theta(h) = \begin{cases} \theta_r + \frac{\theta_s - \theta_r}{(1+(|\alpha h|)^n)^m}, & h < 0 \\ \theta_s, & h \geq 0 \end{cases} \tag{2}$$

130    $$K(h) = K_s S_e^l \left[ 1 - \left( 1 - S_e^{1/m} \right)^m \right]^2 \tag{3}$$

$$m = 1 - 1/n, n > 1 \tag{4}$$

$$S_e = \frac{\theta - \theta_r}{\theta_s - \theta_r} \tag{5}$$

where $\theta_r[L^3 L^{-3}]$ is the residual water content, $\theta_s[L^3 L^{-3}]$ is the saturated water content, $\alpha[L^{-1}]$, $n[-]$ and $m[-]$ are van-Genuchten shape parameters, with the relation given in Eq. (4), $S_e$ [-] is the effective saturation (defined in Eq. (5)) and $l[L]$ is a pore connectivity parameter. The unimodal VGM model was successfully used in several studies to parameterize the hydraulic behavior of variably saturated soils (e.g., Brunetti et al., 2020b; Dettmann et al., 2014; Lambot et al., 2002). It has been shown to become more inconsistent in the clay range of soil textures (Fuentes et al., 1992); however, this limitation does not affect any soils in the framework of this study and was thus employed for all sites. The sink term for the simulation of plant water uptake is implemented according to Eq. (6) (Feddes et al., 1978), where $r_d[L]$ is the root depth, $T_p[L]$ is the potential transpiration and $\alpha(h)$ is a prescribed water stress response function depending on the crop type. The crop parameterization for the sites in this study used the default values for grass cover (Taylor et al., 1972), except for the Pettenbach calibration with maize parameterization according to Wesseling et al. (1991).

$$S(h) = \alpha(h) \frac{1}{r_d} T_p \tag{6}$$

The model domain was set up from soil surface to 1.5 m depth at all sites and two different soil materials were defined for the upper soil (including 20 cm root zone) and the lower soil, respectively. The depths of the soil layers are given in the Appendix in Table A3. The available soil water measurements and profile information (texture data and soil horizons) indicated a distinct topsoil overlying deeper soil layers with low to mild degrees of inhomogeneity at the vast majority of the soil profiles. Dealing with 14 monitoring stations, we uniformly adopted two soil layers with varying thickness across different locations, aiming to reduce the overall computational burden of the Bayesian analysis while maintaining a physically realistic description of the soil domain. Simplifications of the soil profile in the model geometry with a mildly heterogeneous soil will usually lead to an acceptably small loss of accuracy in effective parameters (Schneider et al., 2013).

In this study, we define the point at which percolating water is expected to contribute to groundwater recharge as the amount of water that arrives at the bottom of the area at a depth of 150 cm, well below the root zone. It is assumed that water arriving at this depth will not be subject to further loss mechanisms and so will reach the water table (Heppner et al., 2007). Similar to our approach, Šimůnek (2015) and Heppner et al. (2007) simulated groundwater recharge with HYDRUS-1D for grass-covered soils as bottom flux at 100 cm profile depth; Assefa and Woodbury (2013) used different profile depths of up to 150 cm. However, since the point where water actually reaches the water table remains unknown, the estimation with this approach can be referred to as potential recharge (Scanlon et al., 2002).

Daily time-steps were used in all simulations, for variable boundary conditions as well as simulated soil water content and water fluxes. Meteorological data for the sites, including precipitation, solar radiation, sunshine duration, wind speed,

and relative humidity, were obtained from the Central Institution for Meteorology and Geodynamics (ZAMG), Austria. The potential evapotranspiration $ET_0$ was calculated with the FAO Penman-Monteith method according to Allen et al. (1998). At the upper boundary of the model domain, an "atmospheric", "zero-ponding" boundary condition was specified, where an equilibrium is prescribed between the soil surface pressure and atmospheric water vapor pressure when the evaporative demand exceeds the soil evaporation capacity, and where the pressure at the soil surface is set to zero when both infiltration and surface runoff occur. For the parameter estimation during the half-year calibration periods, as well as for the model validation periods, we chose boundary conditions with respect to the conditions at the measurement plots, i.e. seepage face for the lysimeter sites and free drainage for sites with natural field conditions. For the simulation of long-term potential recharge rates, the lower boundary condition at all sites was set to free-drainage in order to reflect natural conditions with a water table far below the model domain. To improve comparability of long-term simulations at the sites, a grass reference was used with the calibrated Pettenbach model to simulate long-term groundwater recharge. Long-term simulations comprised the entire period of available soil water and meteorological data. For the location Achenkirch, only two years of meteorological data (2017-2018) were available.

For model validation and long-term simulations, snow accumulation and snow melt was accounted for in HYDRUS-1D. The model treats any precipitation falling at a temperature below -2°C as snow and any precipitation above +2°C as liquid, assuming a linear transition between -2°C and +2°C. A 0.4 snow sublimation constant was used for the reduction of potential evaporation from snow and the simulation of snow melt at temperatures above 0°C used a constant of 0.43 cm day $^{-1}$ °C$^{-1}$. This default snow routine in HYDRUS is based on assumptions by Jarvis (1994) and has been found to be suitable for estimating soil water fluxes in unfrozen soils in several studies, (e.g., Assefa and Woodbury, 2013; Zhao et al., 2008).

### 2.2.2 Bayesian analysis

Bayes theorem [Eq. (7)] is the basis for the estimation of parameter posterior distributions which are used for quantification of model parameter uncertainties after calibration.

$$P(\Omega \mid D, M) = \frac{P(D \mid M, \Omega) P(\Omega \mid M)}{P(D \mid M)} \tag{7}$$

Here, $P(\Omega \mid D, M)$ is the posterior probability of the model parameters ($\Omega$), given the data ($D$) and the model ($M$), $P(D \mid M, \Omega)$ is the conditional probability of the data given the model and parameters, $P(\Omega \mid M)$ is the prior probability and $P(D|M)$ is the marginal likelihood or Bayesian model evidence (BME). Prior knowledge, i.e. information available before looking at measured data, is included in the Bayesian inference via the prior distribution which can be chosen as a uniform density bounded by physical limits (e.g., Brunetti et al., 2020b; Gupta et al., 2022; Wöhling et al., 2015). In this study, uniform prior distributions were assumed for all parameters and sites. Their ranges were established based on texture information, literature review, and preliminary testing to prevent truncating posteriors. Final ranges are given in the Appendix in Table A3. By

combining the likelihood and the prior, we obtain a posterior distribution of the most probable SHP values, which reflects the parameters' uncertainty.

We used volumetric water content measurements from TDR sensors in the calibration where the measurement error is based on electromagnetic instantaneous pulses and can be assumed to be independent, homoscedastic, and normally distributed. This leads to a Gaussian likelihood function [Eq. (8)], where $\sigma$ is the standard deviation in the measurement error, $M_i(\Omega)$ is the model realization and $\tilde{y}_i$ is the corresponding observed data.

$$L(\Omega \mid D, M) = \prod_{i=1}^{k} \frac{1}{\sqrt{2\pi\sigma^2}} \exp\left[ -\frac{1}{2}\sigma^{-2} \left( M_i(\Omega) - \tilde{y}_i \right)^2 \right] \tag{8}$$

The choice of likelihood function is critical to the outcome of Bayesian inference and is the subject of ongoing debate. A recent promising approach that should be explored in future studies is the universal likelihood proposed by Vrugt et al. (2022). Instead of making prior assumptions about the distribution of model residuals in the likelihood function, this approach is distribution-adaptive to the actual residual properties. However, in the present study, we used the Gaussian likelihood function as described above for process-based probabilistic inference, where we use significant, systematic discrepancies between model predictions and observations that violate our assumptions as indicators that the model structure needs improvement. We show the residual checks as example for the location Gumpenstein in the Appendix (Fig. A1).

At all 14 locations, 10 soil hydraulic parameters (SHPs) (residual and saturated water content parameters $\theta_r$ and $\theta_s$, shape parameters $\alpha$ and $n$, and the saturated hydraulic conductivity parameter $K_s$, for two soil layers, respectively) were estimated per site. The pore connectivity parameter $l$ was fixed to 0.5 according to Mualem (1976). Together with the SHPs, the standard deviations of the measurement errors were estimated in the Bayesian inference.

The implementation of the Bayesian approach in a numerical framework can become challenging for non-linear models such as the model used here. The Nested Sampling algorithm as proposed by Skilling (2006) has been used successfully for parameter estimation and uncertainty quantification in studies with non-linear hydrological or biogeochemical models (Brunetti et al., 2020a; Elsheikh et al., 2013). It has been tested in Schübl et al. (2022) with synthetic data scenarios for SHP estimation with similar HYDRUS models where it reliably inferred the true parameter values as well as standard deviations of the artificial errors in the calibration data. Nested Sampling is an efficient Monte Carlo method which estimates the Bayesian model evidence and calculates posterior distributions as a side product. It transforms the multi-dimensional integral of the Bayesian model evidence (BME) into a one-dimensional one, which is then solved iteratively, based on the evaluation and redistribution of a number of "live points" over the parameter space. Several improvements were implemented with the original algorithm such as the ellipsoidal rejection sampling scheme which is able to establish multiple posterior modes. This has been realized in the algorithm MULTINEST by Feroz et al. (2009). The algorithm has been shown to be well suited to multimodal distributions and moderately complex inverse problems with up to 20 parameters (Buchner, 2016; Feroz and Hobson, 2008). The algorithm is particularly suitable for our study because it offers a high level of efficiency for unimodal problems while also

handling the possibility of multimodal posteriors. Further details on the algorithm can be found in Feroz et al. (2019, 2009), Feroz and Hobson (2008) and Mukherjee et al. (2006).

Here, we used a number of live points N=100 to sample the parameter space. This number has been shown to produce a reliable estimate of the BME integral (and therefore a satisfactory sampling of the parameter space) in a sensitivity analysis by Brunetti et al. (2020a, b) for similar models and dimensionalities. At each iteration of the algorithm, the current maximum likelihood sample point is multiplied with the remaining prior volume to estimate the maximum remaining volume of the BME integral. Sampling is then terminated according to a tolerance (convergence) criterion, which defines when the remaining

contribution from the current live points to the integral is considered to be small enough. At this point, it is expected, that the bulk of the posterior has been sampled sufficiently. The tolerance parameter in this study was set to 0.5. The number of posterior samples provided by MULTINEST depends on the algorithm convergence with each model. On average, we obtained 4100 posterior samples and corresponding sample weights to characterize posterior parameter distributions. We used 100 random samples from the posterior to propagate parameter uncertainty in the model for long-term simulations to quantify the resulting

uncertainty in recharge simulations. Uncertainty ranges for SHPs and soil water fluxes are given as 95% interquantile ranges (IQR).

### 2.2.3    Statistical analysis

Simulations with the successfully calibrated models were used in a second step to perform a statistical analysis in order to characterize and describe the variability of groundwater recharge at the monitoring sites and to assess the influence of climatic,

geographic and soil properties on potential groundwater recharge rates and their temporal variability. For this purpose, we used a Principle Component Analysis (PCA) and established clusters of sites with similar properties using Agglomerative Clustering (Pedregosa et al., 2011). In order to quantify the temporal variability in water balance components, we calculated the coefficients of variations (CVs) defined as the quotient of standard deviations between months within a year as measure for seasonal variability. Spearman's Rho correlations were used to identify predictor variables for potential groundwater recharge

rates and temporal variability. Significance of correlations were evaluated at a 90% confidence level (p<0.1).

## 3    Results and discussion

### 3.1    Calibration and validation

The required number of iterations of the MULTINEST algorithm with models for all 14 locations ranged between 2595 and 5515 (4111 on average) until the termination criterion was satisfied (as described in Sect. 2.2.2), generally resulting in unimodal

posterior parameter distributions. Median parameter estimates and estimated measurement errors including the 95% credible interval are given in Table 1 for upper and lower soil layers at the 14 sites. Figure 2 shows exemplarily for the location Gumpenstein the calibrated measurement error and median prediction of the volumetric soil water content for the upper and

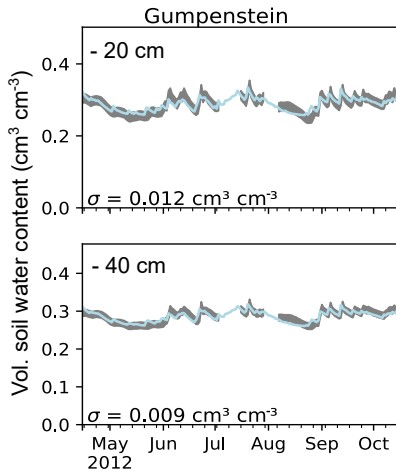

**Figure 2.** Parameter estimation at Gumpenstein: calibration period with soil water content measurements (grey) from two depth levels including the calibrated measurement error $\sigma$, and prediction with median parameter estimates (blue).

lower soil layer. Calibration plots for all 14 sites are shown in Fig. A2 in the Appendix. Uncertainty in the parameter estimation is summarized for all 14 sites in Fig. 3 as ratios between the 95% interquantile range (IQR) and the median estimate.

Median estimates for the VGM shape-parameters $\alpha$ and $n$ varied between $0.001 - 0.945$ cm$^{-1}$ and $1.01 - 2.30$, respectively, where $\alpha$ was $< 0.01$ cm$^{-1}$ at most sites. Except for the high $\alpha$ estimates at Gschlössboden ($\alpha_1$ = 0.945 cm$^{-1}$) and Lobau ($\alpha_1$= 0.511 cm$^{-1}$ and $\alpha_2$ = 0.696 cm$^{-1}$), the VGM shape parameters fell well within the range of values predicted by the ROSETTA pedotransfer model (Schaap and Leij, 1998); high estimates for $\alpha$ and $n$ coincided with a high reported fraction in sand. Median estimates for hydraulic conductivity parameters $K_{\mathrm{s}}$ ranged from 5–3863 cm d$^{-1}$, where high values were found
for soils with high fractions in organic and stone content (Gschlössboden, Sillianberger Alm, Zettersfeld).

**Table 1.** Median estimates and 95 % credible interval of soil hydraulic parameters and measurement errors for upper (L1) and lower (L2) soil profiles.

| Site | | $\theta_r \left( cm^3\ cm^{-3} \right)$ | $\theta_s \left( cm^3\ cm^{-3} \right)$ | $\alpha \left( cm^{-1} \right)$ | $n(-)$ | $K_s \left( cm\ d^{-1} \right)$ | $\sigma_{meas} \left( cm^3\ cm^{-3} \right)$ |
|---|---|---|---|---|---|---|---|
| Lauterach | L1 | $0.134^{+0.062}_{-0.118}$ | $0.425^{+0.019}_{-0.010}$ | $0.002^{+0.002}_{-0.001}$ | $1.34^{+0.19}_{-0.14}$ | $133.9^{+65.2}_{-92.4}$ | $0.024^{+0.002}_{-0.002}$ |
| | L2 | $0.068^{+0.094}_{-0.062}$ | $0.390^{+0.072}_{-0.008}$ | $0.006^{+0.028}_{-0.002}$ | $1.19^{+0.09}_{-0.07}$ | $5.3^{+165.0}_{-1.3}$ | $0.027^{+0.003}_{-0.002}$ |
| Leutasch | L1 | $0.022^{+0.046}_{-0.021}$ | $0.462^{+0.028}_{-0.067}$ | $0.006^{+0.004}_{-0.002}$ | $1.20^{+0.05}_{-0.06}$ | $667.3^{+290.3}_{-339.2}$ | $0.031^{+0.002}_{-0.002}$ |
| | L2 | $0.096^{+0.002}_{-0.003}$ | $0.160^{+0.010}_{-0.009}$ | $0.005^{+0.003}_{-0.002}$ | $2.30^{+0.33}_{-0.27}$ | $770.1^{+219.1}_{-272.0}$ | $0.011^{+0.000}_{-0.001}$ |
| Achenkirch | L1 | $0.023^{+0.054}_{-0.021}$ | $0.570^{+0.025}_{-0.020}$ | $0.001^{+0.001}_{-0.000}$ | $1.13^{+0.01}_{-0.02}$ | $776.3^{+207.2}_{-333.4}$ | $0.048^{+0.003}_{-0.004}$ |
| | L2 | $0.001^{+0.001}_{-0.001}$ | $0.197^{+0.002}_{-0.003}$ | $0.004^{+0.002}_{-0.001}$ | $1.09^{+0.01}_{-0.01}$ | $1843.8^{+1.011.6}_{-708.7}$ | $0.011^{+0.000}_{-0.000}$ |
| Gschlössboden | L1 | $0.050^{+0.000}_{-0.001}$ | $0.278^{+0.047}_{-0.025}$ | $0.945^{+0.053}_{-0.126}$ | $2.25^{+0.27}_{-0.07}$ | $839.0^{+605.4}_{-284.6}$ | $0.021^{+0.001}_{-0.001}$ |
| | L2 | $0.005^{+0.006}_{-0.005}$ | $0.320^{+0.023}_{-0.046}$ | $0.002^{+0.003}_{-0.001}$ | $2.04^{+0.19}_{-0.17}$ | $2320.5^{+293.5}_{-999.0}$ | $0.009^{+0.001}_{-0.001}$ |
| Sillianberger Alm | L1 | $0.143^{+0.052}_{-0.093}$ | $0.536^{+0.042}_{-0.051}$ | $0.006^{+0.009}_{-0.004}$ | $1.12^{+0.03}_{-0.02}$ | $3098.7^{+1769.1}_{-2043.5}$ | $0.030^{+0.002}_{-0.002}$ |
| | L2 | $0.189^{+0.010}_{-0.040}$ | $0.535^{+0.022}_{-0.016}$ | $0.002^{+0.001}_{-0.001}$ | $1.11^{+0.02}_{-0.01}$ | $3863.4^{+1034.2}_{-2520.3}$ | $0.023^{+0.002}_{-0.002}$ |
| Zettersfeld | L1 | $0.082^{+0.061}_{-0.047}$ | $0.583^{+0.015}_{-0.012}$ | $0.060^{+0.228}_{-0.027}$ | $1.09^{+0.02}_{-0.02}$ | $562.0^{+1194.2}_{-290.5}$ | $0.030^{+0.002}_{-0.002}$ |
| | L2 | $0.019^{+0.022}_{-0.017}$ | $0.256^{+0.009}_{-0.010}$ | $0.001^{+0.001}_{-0.000}$ | $1.09^{+0.02}_{-0.01}$ | $3344.6^{+1061.8}_{-1040.8}$ | $0.007^{+0.000}_{-0.001}$ |
| Elsbethen | L1 | $0.105^{+0.082}_{-0.082}$ | $0.453^{+0.012}_{-0.006}$ | $0.001^{+0.001}_{-0.000}$ | $1.13^{+0.05}_{-0.03}$ | $144.8^{+52.7}_{-87.3}$ | $0.013^{+0.001}_{-0.001}$ |
| | L2 | $0.031^{+0.074}_{-0.028}$ | $0.408^{+0.020}_{-0.010}$ | $0.001^{+0.001}_{-0.000}$ | $1.16^{+0.06}_{-0.03}$ | $18.3^{+37.7}_{-8.5}$ | $0.019^{+0.002}_{-0.002}$ |
| Gumpenstein | L1 | $0.051^{+0.027}_{-0.038}$ | $0.375^{+0.014}_{-0.010}$ | $0.003^{+0.001}_{-0.001}$ | $1.08^{+0.01}_{-0.01}$ | $392.1^{+97.6}_{-111.6}$ | $0.012^{+0.001}_{-0.001}$ |
| | L2 | $0.067^{+0.034}_{-0.050}$ | $0.333^{+0.008}_{-0.007}$ | $0.001^{+0.001}_{-0.001}$ | $1.08^{+0.01}_{-0.02}$ | $214.2^{+172.5}_{-105.6}$ | $0.009^{+0.001}_{-0.001}$ |
| Aichfeld- Murboden | L1 | $0.214^{+0.035}_{-0.052}$ | $0.391^{+0.010}_{-0.004}$ | $0.026^{+0.048}_{-0.012}$ | $1.06^{+0.02}_{-0.02}$ | $856.3^{+135.1}_{-294.8}$ | $0.021^{+0.001}_{-0.001}$ |
| | L2 | $0.100^{+0.015}_{-0.015}$ | $0.245^{+0.023}_{-0.017}$ | $0.661^{+0.299}_{-0.264}$ | $1.23^{+0.08}_{-0.05}$ | $57.0^{+77.1}_{-33.8}$ | $0.008^{+0.001}_{-0.000}$ |
| Kalsdorf | L1 | $0.036^{+0.044}_{-0.030}$ | $0.448^{+0.080}_{-0.078}$ | $0.011^{+0.008}_{-0.006}$ | $1.46^{+0.24}_{-0.12}$ | $486.9^{+469.1}_{-367.3}$ | $0.043^{+0.004}_{-0.003}$ |
| | L2 | $0.017^{+0.016}_{-0.016}$ | $0.309^{+0.024}_{-0.009}$ | $0.033^{+0.020}_{-0.011}$ | $1.50^{+0.14}_{-0.08}$ | $867.4^{+130.4}_{-301.3}$ | $0.016^{+0.002}_{-0.001}$ |
| Pettenbach | L1 | $0.063^{+0.108}_{-0.057}$ | $0.387^{+0.005}_{-0.006}$ | $0.001^{+0.001}_{-0.000}$ | $1.15^{+0.06}_{-0.04}$ | $245.3^{+239.7}_{-189.9}$ | $0.036^{+0.004}_{-0.002}$ |
| | L2 | $0.163^{+0.061}_{-0.068}$ | $0.405^{+0.006}_{-0.007}$ | $0.516^{+0.467}_{-0.421}$ | $1.03^{+0.01}_{-0.01}$ | $19.6^{+99.8}_{-16.2}$ | $0.012^{+0.001}_{-0.001}$ |
| Schalladorf | L1 | $0.013^{+0.033}_{-0.012}$ | $0.455^{+0.039}_{-0.034}$ | $0.011^{+0.007}_{-0.005}$ | $1.28^{+0.06}_{-0.05}$ | $17.1^{+27.2}_{-10.7}$ | $0.023^{+0.002}_{-0.001}$ |
| | L2 | $0.049^{+0.066}_{-0.046}$ | $0.395^{+0.005}_{-0.002}$ | $0.001^{+0.001}_{-0.001}$ | $1.22^{+0.06}_{-0.07}$ | $1.5^{+1.2}_{-0.5}$ | $0.005^{+0.001}_{-0.000}$ |
| Lobau | L1 | $0.006^{+0.013}_{-0.006}$ | $0.723^{+0.019}_{-0.032}$ | $0.511^{+0.266}_{-0.106}$ | $1.18^{+0.01}_{-0.01}$ | $684.5^{+201.0}_{-196.4}$ | $0.044^{+0.002}_{-0.002}$ |
| | L2 | $0.173^{+0.052}_{-0.055}$ | $0.378^{+0.004}_{-0.004}$ | $0.696^{+0.268}_{-0.304}$ | $1.01^{+0.01}_{-0.00}$ | $262.0^{+491.5}_{-149.0}$ | $0.004^{+0.000}_{-0.000}$ |
| Frauenkirchen | L1 | $0.049^{+0.045}_{-0.047}$ | $0.489^{+0.052}_{-0.043}$ | $0.001^{+0.001}_{-0.000}$ | $1.46^{+0.12}_{-0.10}$ | $333.4^{+150.2}_{-212.4}$ | $0.0299^{+0.003}_{-0.003}$ |
| | L2 | $0.008^{+0.012}_{-0.007}$ | $0.359^{+0.043}_{-0.031}$ | $0.002^{+0.001}_{-0.000}$ | $1.32^{+0.07}_{-0.03}$ | $269.8^{+195.1}_{-117.5}$ | $0.019^{+0.002}_{-0.001}$ |

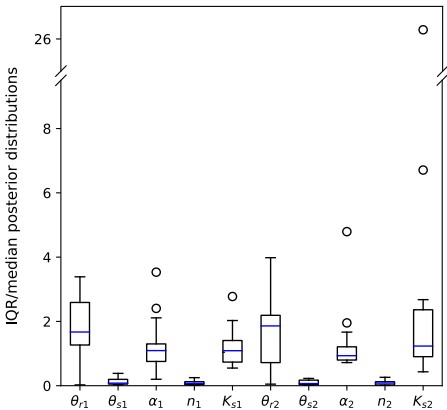

**Figure 3.** Boxplots of estimated parameter uncertainties (index 1 for upper, index 2 for lower soil layer) from all 14 sites, as ratios between 95% interquantile range (IQR) and median estimates.

A typical example for marginal posterior distributions resulting from SHP estimation on the basis of volumetric soil water content data in this study is shown in Fig. 4 for the upper soil layer of the mountain location Zettersfeld. Limits of the plot axes are given by the prior bounds. This representation shows how well the calibration data constrained uncertainties of each parameter: the posterior range of $\theta_r$ is only slightly reduced compared to the prior range, indicating that $\theta_r$ was least sensitive

for simulating soil water content and poorly informed by observations. The parameter $K_s$ has a wide posterior range (although clearly reduced compared to the prior), showing a logarithmic distribution and a clearly defined mode. On the other hand, the parameters $\alpha$, and especially $n$ and $\theta_s$, show narrow posterior distributions which appear leptokurtic, indicating a higher sensitivity for the soil water content simulations and a high information gain from the calibration data.

Parameter interdependencies in the inverse estimation are reflected in the shapes of bivariate contour or scatter plots of

posteriors (see Fig. A3 in the Appendix for a representation of posteriors with closer axes ranges). By random sampling from the posterior, the effect of these correlations is propagated in the uncertainty in the prediction of soil water fluxes. Usually, a negative relation exists between the VGM shape parameters (e.g., Scharnagl et al., 2011; Vrugt et al., 2003; Romano and Santini, 1999). Here, both $\alpha$ and $n$ show narrow posteriors and stray close to the lower physical bounds (0 and 1, respectively). The correlation of posterior samples for $\alpha$ and $K_s$ can be expected to have some effect on the uncertainty in recharge peak

prediction, for which both parameters (but especially $K_s$ under wet conditions) are sensitive (Schübl et al., 2022). This will be further discussed in section 3.2.

Generally, uncertainties in the estimation of the residual water content parameter $\theta_r$ and the saturated hydraulic conductivity parameter $K_s$ for the sites were high, both for the upper and lower soil layers (IQR/median $\sim$ 26 for $K_{s2}$ at Lauterach). The uncertainty in the shape parameter $\alpha$ was medium with a relative uncertainty (IQR/median) < 6 and mostly low absolute values

in estimates and uncertainty ranges. The shape parameter $n$ and the saturated water content parameter $\theta_s$ were identified with the highest precision (IQR/median<0.5).

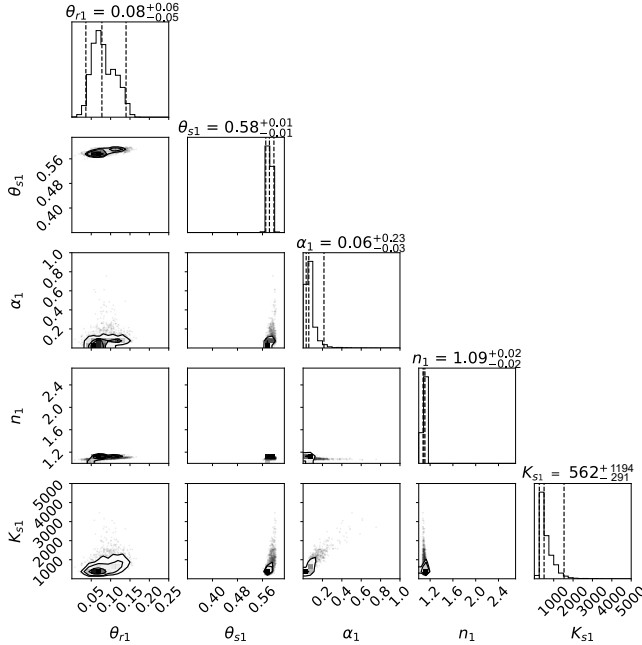

**Figure 4.** Marginal posterior distributions (one-dimensional projection on top of each column; joint distributions of each two parameters below) of estimated SHPs for the top soil layer at Zettersfeld. Presented are residual and saturated water content parameters $\theta_r$ and $\theta_s$ ($cm^3cm^{-3}$), VGM shape parameter $\alpha$ ($cm^{-1}$) and $n$ (-), and the saturated hydraulic conductivity parameter $K_s$ ($cm\ day^{-1}$). Axes ranges correspond to the parameter bounds of the prior distribution. A close up presentation of distributions with narrower axes ranges is shown in Fig. A3 in the Appendix.

Overall, SHP estimation using soil water content monitoring data from different depth levels was associated with some uncertainty. An important factor for parameter uncertainty was soil texture: uncertainties in parameters $K_s$ and $n$, in terms of the 95% interquantile range (IQR) in posteriors, were significantly positively correlated with the percentage in sand (r=0.43 and r=0.42, respectively). Uncertainty ranges in $K_s$, $\alpha$, and $n$ increased significantly with the value of median estimates (Fig. 5). Higher values of these parameters signify a lower water retention capacity of the soil. According to results from Schübl et al. (2022) and Gao et al. (2019), parameter uncertainty from calibration with daily soil water content measurements can be expected to be higher in coarse-textured soils (with a higher soil hydraulic conductivity and lower soil water retention capacity), than in fine-textured soils which was the case in this study. We suppose that the more rapid water flow processes are less efficiently captured in daily soil water content measurements, which are consequently less efficient in constraining uncertainties in SHPs.

We expectedly found high parameter uncertainties for sites where the estimated errors were high ($\sigma > 0.04$ $cm^3\ cm^{-3}$ at Karlsdorf and Lobau) or where the error was high in comparison to the temporal variation (more than 90% of the standard deviation in the observations at Zettersfeld and Sillianberger Alm). Xie et al. (2018) observed how the relation between size

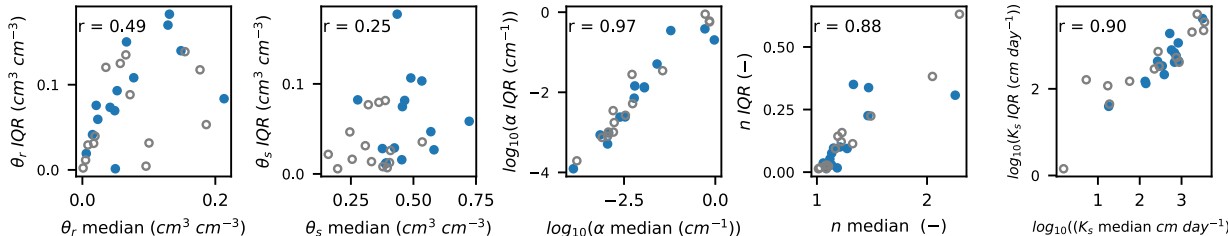

**Figure 5.** Correlations between median parameter estimates and 95% interquantile range (IQR) from posterior parameter distributions for estimated SHPs including 14 sites with each two soil layers (blue dots show the upper soil layer, grey circles show the lower soil layer). Parameters $\alpha$ and $K_s$ are shown on the log-scale to better depict the range of values. Spearman's Rho (r) is given for the presented correlations. They were highly significant with p<0.01 for $\alpha$, $n$, and $K_s$ on the log-scale as well as on the linear scale. On the linear scale, r was slightly lower for $\alpha$ (r=0.77) and $K_s$ (r=0.89).

of the estimated measurement error and temporal variation of the measured variable influences the data's ability to constrain model uncertainties. Brunetti et al. (2019) observed in the estimation of SHPs with remote sensing soil moisture data, that uncertainty in $\theta_r$ estimation was low whereas $\theta_s$ was highly uncertain. This was related to soil water content values being low in their study and mainly representative for unsaturated conditions. In this study, at Lauterach and Elsbethen, very wet climatic conditions and measurements mainly in the wet range resulted in the highest uncertainties in the estimation of $\theta_r$. At Kalsdorf, in contrast, soil moisture dynamics were hardly at saturation and resulted in the highest uncertainty in $\theta_s$ estimation. At the majority of the Austrian locations, soil water content measurements were more often near saturation and less in the dry range (as for example in Fig. 2(a) at Gumpenstein). The $\theta_s$ parameter was therefore mostly better informed by the measurements than $\theta_r$. The estimation of $K_s$ has been frequently shown to be associated with high uncertainties (e.g., Baroni et al., 2010; Minasny and Field, 2005; Mishra et al., 1989).

The reliability of the calibration was quantified by the RMSE between median simulations and observations during calibration and validation periods, summarized for all sites in the Appendix in Table A2. Overall, the calibration fit was good, with RMSE values ranging between 0.009-0.028 cm$^3$ cm$^{-3}$. Some events were missed by the model: at Lauterach and Elsbethen, the drying of the lower soil layer in summer was underestimated; at Gschlössboden, the peak in soil water content in the early calibration period was missed for both layers. For the validation periods, the fit in terms of RMSE deteriorated especially for the locations of Lobau (RMSE calibration = 0.028 cm$^3$ cm$^{-3}$, RMSE validation = 0.054 cm$^3$ cm$^{-3}$) and Pettenbach (RMSE calibration = 0.020 cm$^3$ cm$^{-3}$, RMSE validation = 0.067 cm$^3$ cm$^{-3}$). The Lobau soil profile was under the influence of water table fluctuations where we cannot exclude that model assumptions about the lower boundary condition have been occasionally violated. At the Pettenbach lysimeter station, a crop rotation including fertilization was applied. It is possible, that this affected soil properties, which were assumed to be constant in the modeling. For example, Lu et al. (2020) showed in their review that root growth and decay can alter soil hydraulic properties; Whalley et al. (2005) found, that growing different plants had a

**Table 2.** Local long-term average water balances at 14 sites: Precipitation (P), potential Evapotranspiration ($ET_p$); simulated potential groundwater recharge (GWR) and actual evapotranspiration ($ET_a$) including 95% credible interval from propagated parameter uncertainty.

| | Period | P (mm a$^{-1}$) | $ET_p$(mm a$^{-1}$) | GWR (mm a$^{-1}$) | $ET_a$(mm a$^{-1}$) | GWR/P (%) |
|---|---|---|---|---|---|---|
| Lauterach | $1996-2018$ | 1578 | 700 | $907^{+4}_{-4}$ | $672^{+3}_{-3}$ | $57^{+1}_{-0}\%$ |
| Leutasch | $2008-2018$ | 1235 | 622 | $665^{+9}_{-7}$ | $521^{+7}_{-10}$ | $54^{+2}_{-1}\%$ |
| Achenkirch | $2017-2018$ | 1533 | 673 | $1022^{+14}_{-16}$ | $480^{+14}_{-14}$ | $67^{+1}_{-1}\%$ |
| Gschlössboden | $2012-2018$ | 1493 | 552 | $1319^{+7}_{-9}$ | $170^{+2}_{-6}$ | $88^{+0}_{-1}\%$ |
| Sillianberger Alm | $1997-2018$ | 1023 | 707 | $578^{+13}_{-12}$ | $439^{+10}_{-13}$ | $57^{+1}_{-1}\%$ |
| Zettersfeld | $2012-2018$ | 1353 | 634 | $926^{+15}_{-10}$ | $399^{+10}_{-15}$ | $68^{+1}_{-1}\%$ |
| Elsbethen | $1996-2018$ | 1468 | 665 | $853^{+10}_{-6}$ | $614^{+6}_{-10}$ | $58^{+1}_{-0}\%$ |
| Gumpenstein | $1996-2018$ | 1100 | 661 | $641^{+8}_{-11}$ | $448^{+11}_{-8}$ | $58^{+1}_{-1}\%$ |
| Aichfeld-Murb. | $1996-2018$ | 813 | 728 | $244^{+3}_{-2}$ | $557^{+2}_{-3}$ | $30^{+0}_{-0}\%$ |
| Kalsdorf | $1996-2018$ | 852 | 801 | $229^{+23}_{-24}$ | $623^{+19}_{-31}$ | $27^{+3}_{-3}\%$ |
| Pettenbach | $1996-2018$ | 1031 | 789 | $459^{+18}_{-19}$ | $558^{+20}_{-20}$ | $45^{+2}_{-2}\%$ |
| Schalladorf | $1996-2018$ | 484 | 893 | $45^{+7}_{-7}$ | $431^{+6}_{-7}$ | $9^{+1}_{-1}\%$ |
| Lobau | $1996-2018$ | 570 | 913 | $44^{+8}_{-9}$ | $520^{+9}_{-8}$ | $8^{+1}_{-2}\%$ |
| Frauenkirchen | $2005-2018$ | 601 | 882 | $92^{+15}_{-9}$ | $526^{+10}_{-16}$ | $15^{+2}_{-1}\%$ |

significant effect on the porosity of the soil aggregates, and Schjønning et al. (2002) observed the development different pore systems in soils depending on crop rotation and fertilization.

Overall, in the validation periods RMSE values ranged between 0.014-0.067 cm$^3$ cm$^{-3}$. Scatterplots including the coefficients of determination R$^2$ ($0.34-0.98$) for the validation period are shown in Fig. A4 in the Appendix.

## 3.2 Simulated long-term water balance at the local scale

The calibrated models were used to simulate and assess different components of the water balance for all monitoring stations. In particular, we looked at long-term estimates and temporal variability in actual evapotranspiration and potential groundwater recharge, as well as the average fractions of potential groundwater recharge from precipitation. Long-term averages of input and simulated annual water balance components including propagated parameter uncertainties are given in Table 2. Figure 6 shows cumulative potential recharge sums for the entire simulation period including propagated posterior uncertainty for all 14 sites; Figure 7 shows the uncertainty in the peak prediction as maximum daily recharge rates and posterior uncertainty during the same period.

Uncertainty in the estimated long-term potential annual recharge from propagated parameter uncertainty was highest in Kalsdorf (95% IQR = 47 mm) and lowest in Aichfeld-Murboden (95% IQR = 5 mm). Uncertainties in the prediction of long-term and cumulative recharge rates (Fig. 6) were generally small in relation to the high sums estimated for mountain and

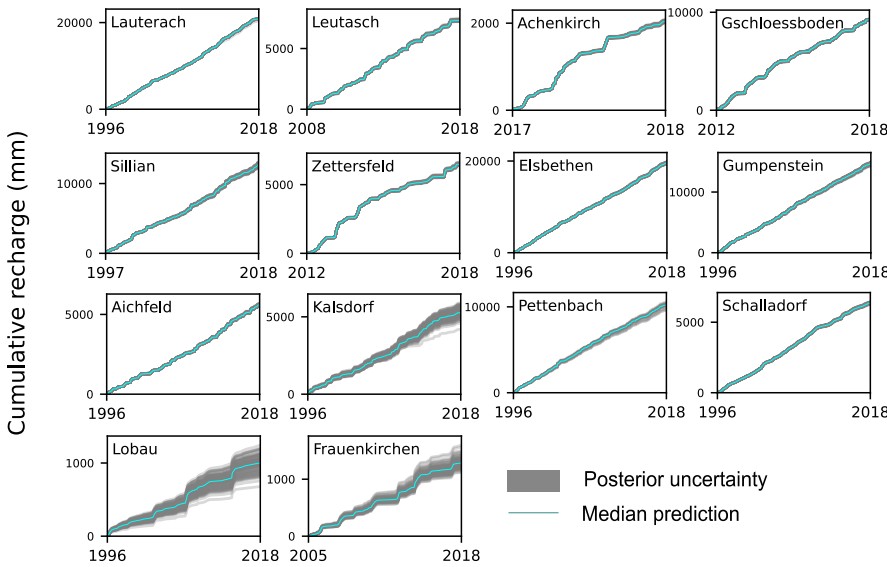

**Figure 6.** Median prediction and posterior uncertainty in the long-term estimation of cumulative recharge at 14 Austrian sites.

western sites; however, in relation to low absolute values at dry eastern sites Kalsdorf, Lobau and Frauenkirchen, posterior uncertainties played a more important role. The relative uncertainty (IQR/median) in long-term recharge estimates ranged between 1% (Gschlössboden, Lauterach) and 39% (Lobau). The prediction of peaks in recharge was generally affected by higher uncertainties (Fig. 7), especially at western mountainous sites with high maximum rates (Lauterach, Leutasch, Gschlössboden, Sillianberger Alm). In a previous study with similar models and hydrological conditions, we found $n$ to be the most sensitive parameter for cumulative recharge prediction, and $K_s$ to be most sensitive for peak prediction, especially under wet climatic conditions (Schübl et al., 2022). Small uncertainties in the prediction of long-term recharge sums here were related to the generally small uncertainties in the VGM shape parameter $n$, whereas higher uncertainties in the hydraulic conductivity parameter $K_s$ (sometimes in interaction with uncertainties in $\alpha$) can be considered the main reason for the greater uncertainty in the peak prediction.

It has often been found that *in-situ* field measurements of soil water content are not sufficient for the accurate and precise estimation of SHPs (e.g., Scharnagl et al., 2011; Ritter et al., 2003). Here, we found that while SHPs were partially affected by considerable uncertainties, the precision was still in an acceptable range for groundwater recharge estimation. At dry locations with small absolute recharge rates, model uncertainties could be further reduced e.g., by including additional observations in the calibration. Especially the combination with soil matric potential measurements has been shown to be highly informative for SHP estimation and to considerably reduce uncertainties in recharge estimation (Schübl et al., 2022; Schelle et al., 2012). Including additional measurements in the analysis, however, might not only lead to different shapes in SHP posteriors, but to altogether different estimates. This issue requires further investigation with available soil water monitoring data.

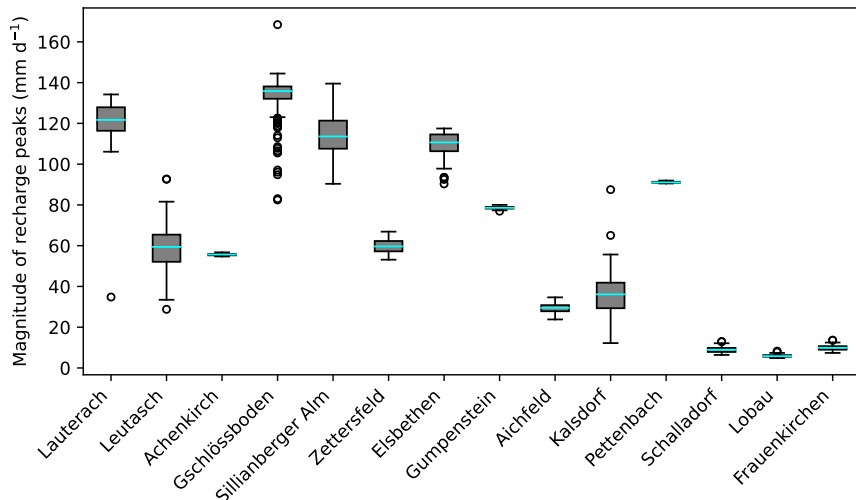

**Figure 7.** Uncertainty ranges in peaks of the potential recharge flux at 14 Austrian sites. Maximum daily rates of the long-term simulation period and propagated posterior uncertainties are presented as boxplots.

Propagated uncertainties in soil water fluxes presented here are a result of parameter uncertainties from the calibration, and the sensitivity of the simulated water fluxes towards the parameters. Uncertainties in water fluxes were treated as aleatory, derived from stationary statistical characteristics. In addition, the epistemic uncertainty associated with the lack of knowledge about the correct representation of system dynamics (conceptual uncertainty) and forcing data may affect the overall predictive uncertainty and reduce the effective information content of observations (Beven, 2016). Due to their complex and often dynamic nature, epistemic uncertainties pose important conceptual and numerical challenges. For instance, model conceptual uncertainty can be assessed by comparing different model structures using specific statistical metrics (e.g., marginal likelihood). This was, however, beyond the scope of the present study, which focuses on the inverse estimation of soil water fluxes at multiple monitoring stations to discuss implications on the water balance. In this framework, an appraisal of the model structural adequacy through posterior predictive checks appears sufficient. We were not able to account for some processes in this study which may have affected water balances at the sites: The modeling approach assumed that the groundwater table was well below the model domain at all times. At the Lobau site, however, the groundwater table is shallow, and fluctuations may have reached into the model domain. In this case, infiltrating water may have reached the water table earlier than assumed by the model. At the same time, net recharge would have been reduced if the capillary fringe extended into the root zone or even to the soil surface and transpiration and evaporation occurred directly from groundwater (Doble and Crosbie, 2017). Further, the modeling approach here neglected preferential and lateral flow processes. The ground surface at the measurement locations was even; however, it has been shown that heterogeneity and layering in the soil profiles can lead to lateral flow, even when the effective hydraulic gradient is vertical (Heilig et al., 2003; Rimon et al., 2007).

To assess the plausibility of estimated potential recharge rates we compared them to literature values where available. Tóth et al. (2016) assumed annual groundwater recharge for the western Pannonian Basin of 70 mm a$^{-1}$. The region includes the three southeasternmost sites here (Lobau, Frauenkirchen and Kalsdorf), where potential recharge rates in this study ranged

between 44 – 229 mm a$^{-1}$. For Wagna in southern Styria, 20 km from Kalsdorf, between 296 – 396 mm a$^{-1}$ have been estimated in studies by Collenteur et al. (2021) and Stumpp et al. (2009). We also compared the estimates with the long-term (1961-1990) water balance averages for precipitation, potential and actual evapotranspiration on the catchment scale from the Hydrological Atlas of Austria (HAO) (BMLFUW, 2007; Dobesch, 2007; Kling et al., 2007b, a) (Fig. A5 in the Appendix). The mean annual areal actual evapotranspiration estimates of the HAO (Kling et al., 2007a) are based on water balance calculations

from the period 1961 to 1990. They are comparable to our long-term estimates (R$^2$ = 0.78) supporting the plausibility of the here established water balances.

We further evaluated estimated recharge rates at the locations of Leutasch and Gumpenstein by comparing the available lysimeter outflow measurements to modeled median estimates. It resulted in an acceptable agreement with R$^2$ = 0.56 (for the period 2008 – 2018) and R$^2$ = 0.64 (for the period 2001 – 2018), respectively, and is shown in Fig. A6 in the Appendix,

including uncertainties. Variability in annual seepage measurements between four Gumpenstein lysimeters was high with an average uncertainty range of 132 mm a$^{-1}$. This clearly exceeded the average range of predictive uncertainty related to parameter uncertainty of the modeling at this site (20 mm a$^{-1}$). Besides the uncertainty in the seepage measurement, the variability in the measurements could also be an indicator of spatial heterogeneities causing differences in the soil hydrology for individual lysimeters. In any case, the high variability in seepage measurements here emphasizes the need to analyze

uncertainties in the estimation of soil water fluxes.

## 3.3   Statistical analysis of hydrologically relevant properties

In the following section we characterize the 14 monitoring according to hydrologically relevant properties including model estimations from the previous section. Since uncertainty in long-term actual evapotranspiration and recharge rates were generally low, and to enable the analysis with common statistical tools, we will proceed from here on using the median values without

consideration of uncertainty ranges.

The seasonal variability in groundwater recharge (quantified as coefficient of variation from standard deviation between monthly sums and annual means) ranged between 71% and 265%. This was consistently higher than the seasonality in precipitation (52 – 76%) and potential evapotranspiration (64 – 76%) indicating that potential recharge rates vary significantly more over the year than the meteorological input variables. We further analyzed the seasonality in local water balances in a

PCA and correlation analysis. Figure 8 shows the biplot of the PCA with first and second principle components (PC1 and PC2, explaining 77% of the variance in the data), according to amount and seasonality of water balance components, the fraction of potential groundwater recharge from precipitation, and site specific properties (altitude and longitude; sand, silt, clay and organic matter percentages of the upper soil layers).

Two clusters were established: The five sites in the south and east of Austria (Aichfeld (9), Kalsdorf (10), Schalladorf (12),

Lobau (13), Frauenkirchen (14)) show a potential recharge fraction of less than 30% of annual precipitation (as low as 8% in

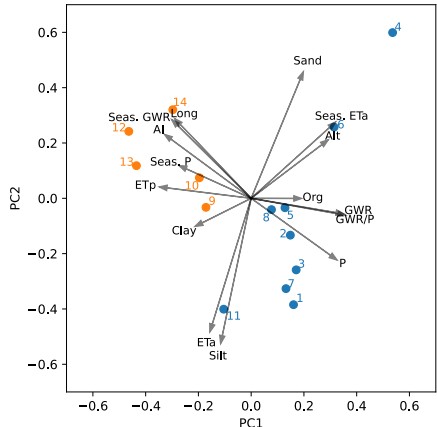

**Figure 8.** Principle Component Analysis Biplot. Included variables are potential annual groundwater recharge (GWR), annual precipitation (P), annual potential evapotranspiration ($ET_p$), annual actual evapotranspiration ($ET_a$), the fraction of groundwater from precipitation (GWR/P), seasonalities (Season.) in GWR, P, $ET_a$; longitude (Long), altitude (Alt); sand, silt, clay and organic matter (Org) percentages at the 14 sites. Clusters of monitoring sites with similar characteristics are shown in orange and blue. The clustering with Euclidean affinity and ward linkage, as well as the Biplot were produced using the *sklearn* module in Python by Pedregosa et al. (2011).

Lobau), a high seasonality in groundwater recharge (134 – 265%) and precipitation (67 – 76%), but a low seasonality in actual evapotranspiration (59 – 73%). The remaining nine out of 14 sites in western to central Austria with humid to wet climate show a fraction of potential groundwater recharge from precipitation of more than 40%, and a low seasonality in precipitation (52 – 68%). The seasonality in groundwater recharge at these sites was lower than in the East (71 – 124%), but seasonality in

actual evapotranspiration was higher (75 – 112%); it was most pronounced at the three sub-alpine sites (Gschlössboden (4), Sillianberger Alm (5), and Zettersfeld (6)) which were influenced by snow and where little to no actual evapotranspiration was estimated outside of the extended summer period (May – September). An obvious outlier among the monitoring sites in Fig. 8(a) was the location Gschlössboden (4) at high altitude, with coarse soil, lowest potential and actual evapotranspiration, and the highest estimated potential recharge rates compared to other sites.

Figure 9 shows the pair-wise scatterplots, correlation coefficients and significance levels of relevant variables. Since precipitation and potential evapotranspiration were negatively correlated, we adopted the Aridity Index ($ET_p$/P) as predictor instead of looking at both variables separately. Seasonality in potential evapotranspiration is not shown, since no significant correlations to other variables were identified. Grain size classes of the soil textures were intercorrelated, we therefore only used the sand fraction as predictor variable.

Potential annual groundwater recharge rates were negatively correlated with aridity (lower precipitation and higher potential evapotranspiration). This was expected and was also supported by findings of Moeck et al. (2020) on the global scale. At the Austrian sites, aridity increased and potential groundwater recharge decreased significantly with longitude, resulting in lower potential recharge rates at the eastern than at the western sites. Precipitation and recharge rates were higher in the West than in

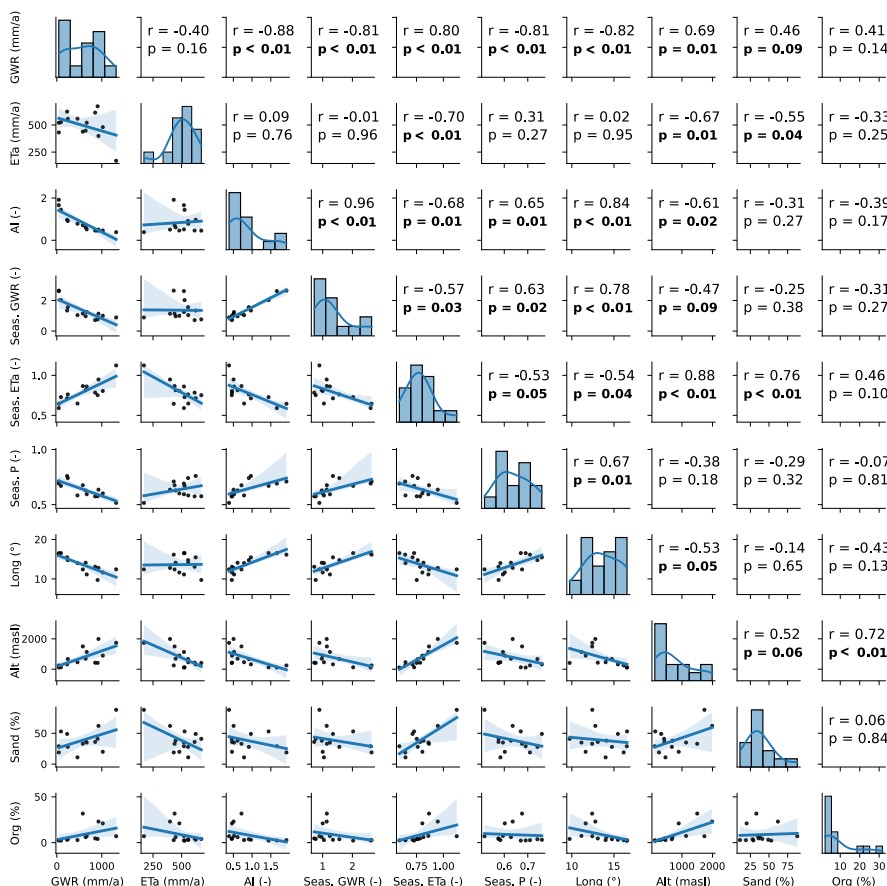

**Figure 9.** Correlation analysis with pair-wise scatter plots, Spearman's Rho correlation coefficient and significance levels for the variables potential annual groundwater recharge (GWR), annual actual evapotransiration (ET$_a$), Aridity Index (AI), Seasonalities (Seas.) in GWR, ET$_a$, and P; longitude (Long), altitude (Alt), percentages in sand and organic matter (Org) at 14 monitoring sites.

the East, following both the longitudinal gradient in altitude and the climatic influence of the wet oceanic climate in the West,
with high precipitation and recharge rates even at lower altitudes (Lauterach, Elsbethen), versus the dry continental climate in the East. In the study here, slopes were not taken into consideration, as the monitoring sites were horizontally even and the modeling domain was limited to the plot scale. Regarding the larger scale (and actual recharge rates), the occurrence of steep slopes at high altitudes would be expected to result in more surface runoff or more interflow instead of recharge (Brunetti et al., 2022; Moeck et al., 2020) which could reverse the correlation of recharge rates with altitude.

The fraction of potential groundwater recharge to precipitation (GWR/P) was strongly correlated with the amount of precipitation (r = 0.91, p<0.001). Similarly, Barron et al. (2012) found an exponential relationship between annual recharge and rainfall estimates at Australian sites, which they explained by the correlation of high amounts of precipitation with high rainfall intensities and long wet periods throughout the year, leading to an increased fraction of recharge from precipitation.

Higher potential recharge rates and lower actual evapotranspiration were correlated with a higher percentage in sand. Soils with greater sand fraction and less fine material have a higher hydraulic conductivity and a lower water retention capacity as they let water percolate faster below the root zone (Emerson, 1995; Wohling et al., 2012). Wang et al. (2009) observed how the fraction of recharge from precipitation increased with coarser soil texture as the more rapid deep percolation reduced evapotranspiration. In the study here, however, the relation between potential groundwater recharge and soil texture was weaker compared to climatic factors, i.e. precipitation and potential evapotranspiration. This corresponded to findings of the global scale analysis by Moeck et al. (2020).

Seasonality in potential groundwater recharge was most strongly correlated with the Aridity Index ($\mathrm{ET_p}/\mathrm{P}$). Sites in the east, with more pronounced aridity and low potential recharge rates, were associated with a high seasonality with extended periods of zero recharge. Estimated potential groundwater recharge there was concentrated on the winter half-year. High rates in potential groundwater recharge were associated with sites where recharge occurred throughout the year and were thus correlated with a low seasonality in recharge. Soil texture did not correlate with seasonality in estimated potential groundwater recharge. In this study, we assumed the same lower boundary for all profiles to ensure comparability of the sites, where additional data from below 1.5 m were not available. However, the depth of the water table, and thus the thickness of the unsaturated zone, in addition to structural features causing lateral flow, determine quantity and timing of water actually reaching the aquifer. With greater thickness of the unsaturated zone, the influence of soil water retention characteristics on magnitude and temporal variability of actual groundwater recharge rates might increase (Burri et al., 2019; Cao et al., 2016; Moeck et al., 2020). In future, data from the deeper unsaturated zone (>1.5 m) would be helpful to further improve the quantification of recharge.

## 4 Conclusions

In this study, we made use of volumetric soil water content measurements from multiple depth levels at 14 locations in Austria to inversely estimate effective soil hydraulic parameters (SHPs) using the physically based HYDRUS-1D model, and we quantified parameter uncertainties in a Bayesian probabilistic framework based on multimodal Nested Sampling. We used the calibrated models for the long-term simulation of soil water fluxes and associated uncertainties. Finally, we compared potential recharge rates and actual evapotranspiration at the 14 Austrian locations to identify the influencing factors on amount and temporal variability of local water balances.

SHPs were successfully established and resulted in adequate fits of model simulations to observations. The parameter estimation based on soil water content measurements was partly subject to considerable uncertainties, especially in residual water content ($\theta_\mathrm{r}$) and soil hydraulic conductivity parameters ($K_\mathrm{s}$). The latter resulted in considerable uncertainties in predicting the magnitude of recharge peaks at the sites. Higher uncertainties in VGM shape parameters $\alpha$, $n$, and soil hydraulic conductivity parameter $K_\mathrm{s}$ were associated with coarser soil textures. In general, however, uncertainty in the estimation of the VGM shape parameters was low and resulted in small uncertainty ranges for long-term potential groundwater recharge rates. Absolute uncertainty ranges were between 5-47 mm $\mathrm{a}^{-1}$, which corresponded to relative uncertainties in cumulative recharge prediction (IQR/median) between 1%, at sites with high absolute rates in a wet climate, and 39% at dry eastern sites with small potential

recharge rates. Especially at the latter sites, model uncertainties could be improved by including additional observations in the calibration.

Estimated potential groundwater recharge rates at the Austrian soil water monitoring sites were influenced by the East-West gradient in altitude and climatic conditions: The dry continental climate at the eastern locations was associated with low fractions of potential groundwater recharge from precipitation, and high seasonality in potential recharge rates. In contrast, the wet and snow influenced climate at western and central Austrian sites came with high potential recharge rates and lower temporal variability in recharge than in the East, but with a higher seasonality in actual evapotranspiration. Sandy soil textures were associated with higher potential recharge rates and lower actual evapotranspiration. However, precipitation and potential evapotranspiration were more influential variables than soil properties on estimated potential recharge rates and their temporal variability.

The approach could be improved by including information on the deeper vadose zone to obtain more insight on temporal variation and seasonality of actual recharge, and to improve the model structure including lower boundary conditions. Especially at dry locations, using improved and additional measurements (e.g. of soil matric potential) could help reduce uncertainty in cumulative recharge estimation. Additionally, consideration of sites with varying slopes and the inclusion of surface runoff simulations in the analysis might improve representativeness for larger scale.

Overall, the use of a Nested Sampling based Bayesian approach proved to be an efficient method to inversely estimate SHPs and soil water fluxes, and to quantify associated uncertainties from soil water monitoring data. The calibrated models can be used to estimate future groundwater recharge rates under climate change and to illuminate model uncertainties resulting from SHP uncertainties and a range of climate scenarios.

*Author contributions.* MS, GB and CS designed the study, MS and GB performed the model simulations and contributed Python code for the analysis and data visualization. MS conducted the statistical analysis. GF curated and provided the original data. MS wrote the initial draft, CS, GB and GF reviewed and edited the manuscript. All authors revised the paper and agreed on its contents.

*Competing interests.* The authors declare that they have no conflict of interest.

*Acknowledgements.* This work was carried out in the framework of the RechAUT project and supported by the Austrian Academy of Sciences (ÖAW), Vienna, Austria. Meteorological data was provided by the Central Institution for Meteorology and Geodynamics (ZAMG), Vienna, Austria. Soil texture data for the site of Gschlössboden was provided by the Hydrography and Hydrology Division of the Tyrolean Government.

# Appendix A

**Table A1.** Site properties and particle size distribution of the upper soil layer (ÖNORM L 1050).

|  | Altitude (m.a.s.l.) | Longitude (°) | Latitude (°) | Sand % $0.063 - 2.0$ mm | Silt % $0.002 - 0.063$ mm | Clay % $< 0.002$ mm |
|---|---|---|---|---|---|---|
| Lauterach | 415 | 9.74 | 47.48 | 41 | 45 | 14 |
| Leutasch | 1135 | 11.14 | 47.37 | 35 | 51 | 14 |
| Achenkirch | 895 | 11.64 | 47.58 | 20 | 48 | 32 |
| Gschlössboden | 1737 | 12.43 | 47.12 | 88 | 12 | 0 |
| Sillianberger Alm | 1500 | 12.41 | 46.76 | 33 | 63 | 4 |
| Zettersfeld | 1990 | 12.79 | 46.87 | 56 | 42 | 2 |
| Elsbethen | 428 | 13.08 | 47.76 | 36 | 59 | 5 |
| Gumpenstein | 690 | 14.10 | 47.50 | 38 | 53 | 9 |
| Aichfeld-Murb. | 669 | 14.76 | 47.21 | 28 | 56 | 16 |
| Kalsdorf | 320 | 15.47 | 46.95 | 49 | 42 | 9 |
| Pettenbach | 466 | 14.01 | 47.98 | 11 | 75 | 14 |
| Schalladorf | 238 | 16.14 | 48.64 | 17 | 43 | 40 |
| Lobau | 150 | 16.53 | 48.21 | 29 | 57 | 14 |
| Frauenkirchen | 124 | 16.90 | 47.85 | 53 | 33 | 14 |

**Table A2.** Calibration and validation periods, and goodness of fit (root mean squared error RMSE) between median prediction and measurements.

| | Calibration | Validation | Calib. RMSE $(\text{cm}^3\ \text{cm}^{-3})$ | Valid. RMSE $(\text{cm}^3\ \text{cm}^{-3})$ |
|---|---|---|---|---|
| Lauterach | $01.03. - 31.10.2015$ | $01.01.2016 - 31.12.2016$ | 0.025 | 0.028 |
| Leutasch | $01.03. - 31.10.2014$ | $01.03.2017 - 31.10.2017$ | 0.018 | 0.021 |
| Achenkirch | $01.05. - 31.10.2018$ | $01.01.2017 - 31.12.2017$ | 0.023 | 0.037 |
| Gschlössboden* | $01.04. - 30.09.2018$ | $01.01.2018 - 31.12.2018$ | 0.017 | 0.019 |
| Sillianberger Alm* | $01.03. - 31.10.2018$ | $01.01.2018 - 31.12.2018$ | 0.026 | 0.020 |
| Zettersfeld | $01.04. - 30.09.2017$ | $01.01.2014 - 31.12.2015$ | 0.022 | 0.020 |
| Elsbethen | $01.03. - 31.10.2015$ | $01.01.2012 - 31.12.2012$ | 0.018 | 0.015 |
| Gumpenstein | $15.04. - 15.10.2012$ | $01.03.2011 - 31.12.2011$ | 0.011 | 0.014 |
| Aichfeld-Murb. | $15.04. - 15.10.2016$ | $15.08.2017 - 31.12.2018$ | 0.015 | 0.021 |
| Kalsdorf | $01.03. - 31.10.2007$ | $01.01.2008 - 31.12.2008$ | 0.021 | 0.037 |
| Pettenbach** | $23.04. - 14.10.2014$ | $24.04.2013 - 24.09.2013$ | 0.020 | 0.067 |
| Schalladorf | $01.03. - 31.10.2010$ | $01.03.2013 - 31.10.2014$ | 0.009 | 0.028 |
| Lobau | $01.03. - 31.10.2012$ | $01.01.2000 - 31.12.2000$ | 0.028 | 0.054 |
| Frauenkirchen | $01.03. - 31.10.2015$ | $01.01.2012 - 31.12.2014$ | 0.021 | 0.036 |

\* No validation data available outside the calibration year, instead the RMSE for the entire year (2018) was calculated.

\*\* Pettenbach calibration period during maize cultivation, validation period during soy bean cultivation. Root parameters were adjusted and potential evapotranspiration estimation was estimated with corresponding crop coefficients (Allen et al., 1998).

**Table A3.** Soil layers in HYDRUS-1D and prior parameter ranges of the Bayesian analysis.

| Site | | Depth (cm) | $\theta_r \left( \mathrm{cm^3\ cm^{-3}} \right)$ | $\theta_s \left( \mathrm{cm^3\ cm^{-3}} \right)$ | $\alpha \left( \mathrm{cm^{-1}} \right)$ | $n(-)$ | $K_s \left( \mathrm{cm\ d^{-1}} \right)$ |
|---|---|---|---|---|---|---|---|
| Lauterach | L1 | $0 - 24$ | $0.00 - 0.20$ | $0.30 - 0.50$ | $0.0001 - 0.1000$ | $1.01 - 2.00$ | $1 - 200$ |
| | L2 | $25 - 150$ | $0.00 - 0.20$ | $0.30 - 0.50$ | $0.0001 - 0.1000$ | $1.01 - 2.00$ | $1 - 200$ |
| Leutasch | L1 | $0 - 24$ | $0.00 - 0.10$ | $0.25 - 0.50$ | $0.0001 - 0.5000$ | $1.01 - 2.70$ | $1 - 1000$ |
| | L2 | $25 - 150$ | $0.00 - 0.10$ | $0.15 - 0.40$ | $0.0001 - 0.5000$ | $1.01 - 3.50$ | $1 - 1000$ |
| Achenkirch | L1 | $0 - 15$ | $0.00 - 0.25$ | $0.40 - 0.60$ | $0.0001 - 0.5000$ | $1.01 - 2.70$ | $1 - 1000$ |
| | L2 | $16 - 150$ | $0.00 - 0.08$ | $0.10 - 0.20$ | $0.0001 - 1.0000$ | $1.01 - 3.50$ | $10 - 10000$ |
| Gschlössboden | L1 | $0 - 22$ | $0.00 - 0.05$ | $0.20 - 0.35$ | $0.0001 - 1.0000$ | $1.01 - 2.70$ | $10 - 10000$ |
| | L2 | $23 - 150$ | $0.00 - 0.05$ | $0.20 - 0.35$ | $0.0001 - 1.0000$ | $1.01 - 3.50$ | $10 - 10000$ |
| Sillianberger Alm | L1 | $0 - 24$ | $0.00 - 0.20$ | $0.30 - 0.60$ | $0.0001 - 0.2000$ | $1.01 - 2.00$ | $1 - 5000$ |
| | L2 | $25 - 150$ | $0.00 - 0.20$ | $0.30 - 0.60$ | $0.0001 - 0.2000$ | $1.01 - 2.00$ | $1 - 5000$ |
| Zettersfeld | L1 | $0 - 49$ | $0.00 - 0.25$ | $0.30 - 0.60$ | $0.0001 - 1.0000$ | $1.01 - 2.70$ | $1 - 5000$ |
| | L2 | $50 - 150$ | $0.00 - 0.08$ | $0.10 - 0.40$ | $0.0001 - 1.0000$ | $1.01 - 3.50$ | $1 - 5000$ |
| Elsbethen | L1 | $0 - 24$ | $0.00 - 0.20$ | $0.30 - 0.50$ | $0.0001 - 0.1000$ | $1.01 - 2.00$ | $1 - 200$ |
| | L2 | $25 - 150$ | $0.00 - 0.20$ | $0.30 - 0.50$ | $0.0001 - 0.1000$ | $1.01 - 2.00$ | $1 - 200$ |
| Gumpenstein | L1 | $0 - 24$ | $0.00 - 0.20$ | $0.25 - 0.60$ | $0.0001 - 0.5000$ | $1.01 - 2.70$ | $0.1 - 500$ |
| | L2 | $25 - 150$ | $0.00 - 0.20$ | $0.25 - 0.60$ | $0.0001 - 0.5000$ | $1.01 - 2.70$ | $0.1 - 500$ |
| Aichfeld-Murboden | L1 | $0 - 74$ | $0.00 - 0.25$ | $0.30 - 0.60$ | $0.0001 - 0.5000$ | $1.01 - 2.70$ | $1 - 1000$ |
| | L2 | $75 - 150$ | $0.00 - 0.15$ | $0.17 - 0.40$ | $0.0001 - 1.0000$ | $1.01 - 2.70$ | $1 - 1000$ |
| Kalsdorf | L1 | $0 - 24$ | $0.00 - 0.10$ | $0.30 - 0.60$ | $0.0001 - 0.2000$ | $1.01 - 2.00$ | $1 - 1000$ |
| | L2 | $25 - 150$ | $0.00 - 0.10$ | $0.30 - 0.60$ | $0.0001 - 0.2000$ | $1.01 - 2.00$ | $1 - 1000$ |
| Pettenbach | L1 | $0 - 24$ | $0.00 - 0.25$ | $0.30 - 0.60$ | $0.0001 - 0.5000$ | $1.01 - 2.70$ | $0.1 - 500$ |
| | L2 | $25 - 150$ | $0.00 - 0.25$ | $0.30 - 0.60$ | $0.0001 - 1.0000$ | $1.01 - 2.70$ | $0.1 - 500$ |
| Schalladorf | L1 | $0 - 44$ | $0.00 - 0.20$ | $0.40 - 0.60$ | $0.0001 - 0.1000$ | $1.01 - 2.00$ | $1 - 50$ |
| | L2 | $45 - 150$ | $0.00 - 0.20$ | $0.30 - 0.50$ | $0.0001 - 0.1000$ | $1.01 - 2.00$ | $1 - 50$ |
| Lobau | L1 | $0 - 100$ | $0.00 - 0.15$ | $0.35 - 0.75$ | $0.0001 - 1.0000$ | $1.01 - 2.70$ | $1 - 1000$ |
| | L2 | $101 - 150$ | $0.00 - 0.25$ | $0.35 - 0.60$ | $0.0001 - 1.0000$ | $1.01 - 2.70$ | $1 - 1000$ |
| Frauenkirchen | L1 | $0 - 24$ | $0.00 - 0.20$ | $0.30 - 0.60$ | $0.0001 - 0.2000$ | $1.01 - 2.00$ | $1 - 500$ |
| | L2 | $25 - 150$ | $0.00 - 0.20$ | $0.30 - 0.60$ | $0.0001 - 0.2000$ | $1.01 - 2.00$ | $1 - 500$ |

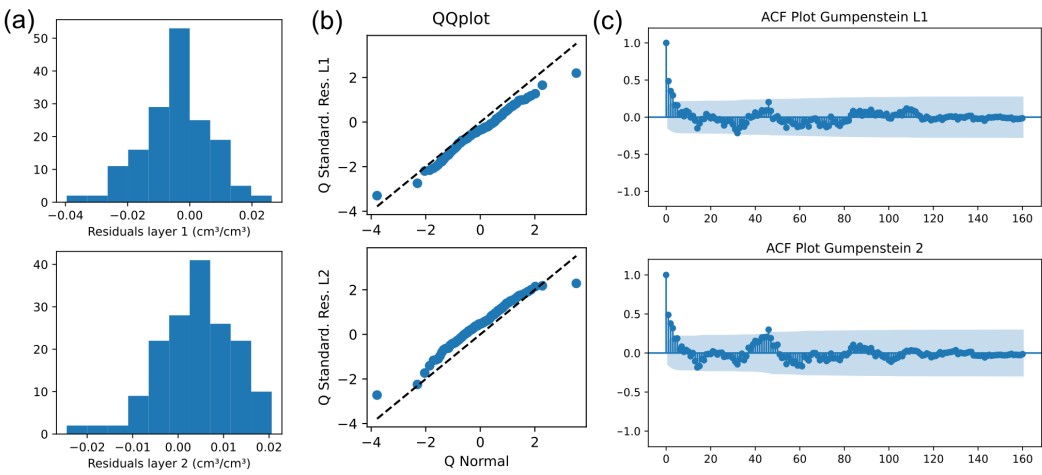

**Figure A1.** Residual plots for the calibration at Gumpenstein: (a) histogram of residuals, (b) quantile-quantile (QQ) plots and (c) autocorrelation function (ACF) plots. The upper graphs show residuals of the top soil layer (L1), the lower graphs show residuals of the lower soil layer (L2).

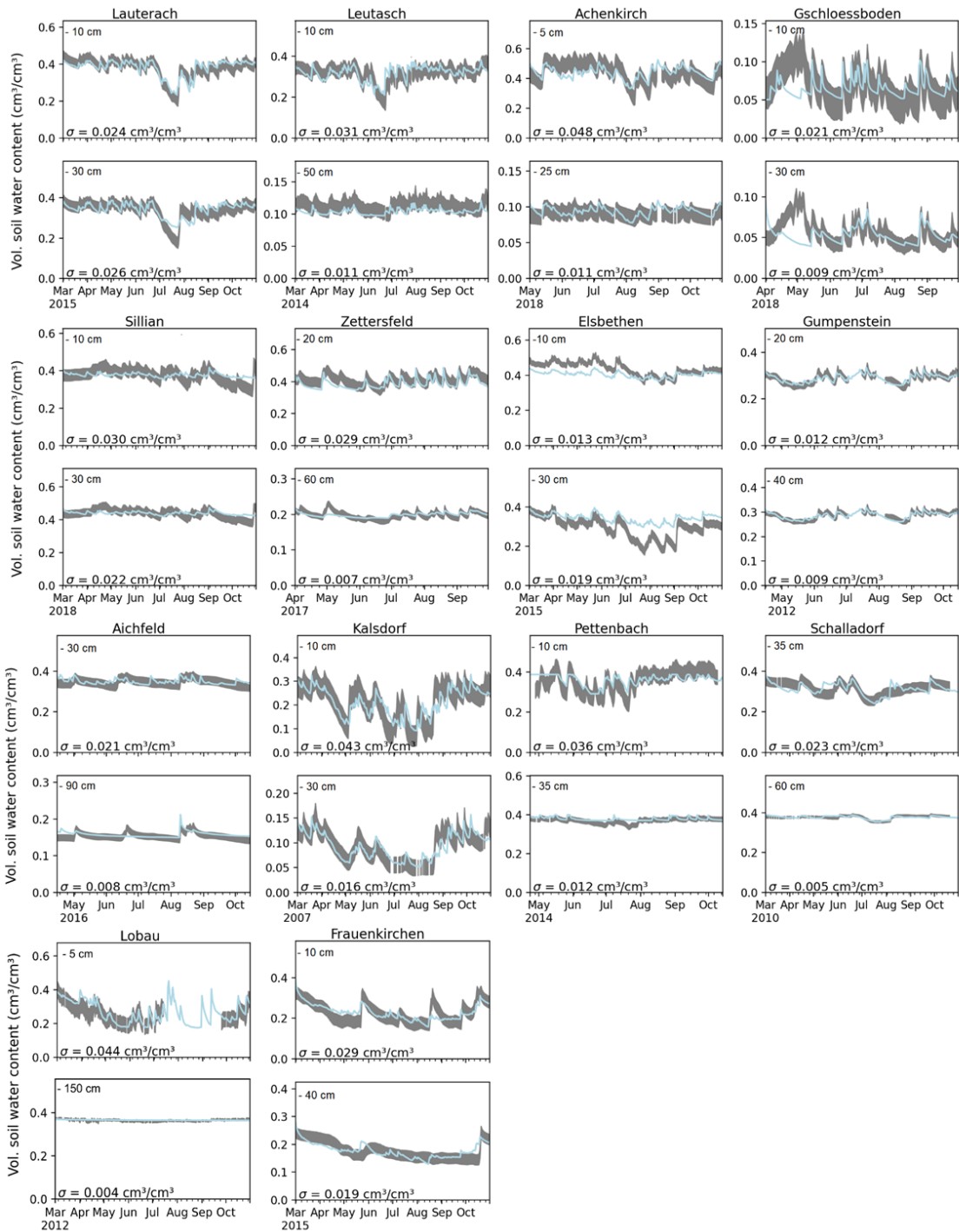

**Figure A2.** Calibration with soil water content measurements at all 14 sites: The grey bands show the measurement including the area of the calibrated measurement error $\sigma$, the blue lines show the prediction with median parameter estimates for each one measurement depth in upper and lower soil layer.

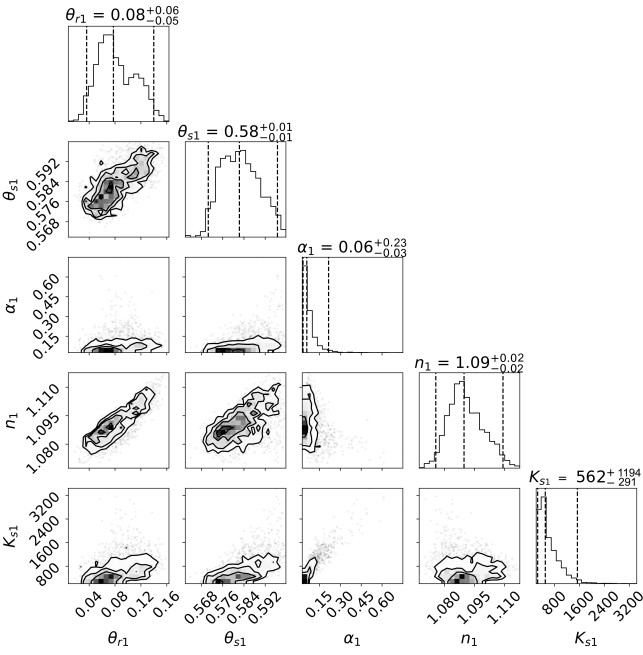

**Figure A3.** Marginal posterior distributions (one-dimensional projection on top of each column; joint distributions of each two parameters below) of estimated SHPs for the top soil layer at Zettersfeld. Presented are residual and saturated water content parameters $\theta_r$ and $\theta_s$ ($cm^3 cm^{-3}$), VGM shape parameter $\alpha$ ($cm^{-1}$) and $n$ (-), and the saturated hydraulic conductivity parameter $K_s$ ($cm\ day^{-1}$).

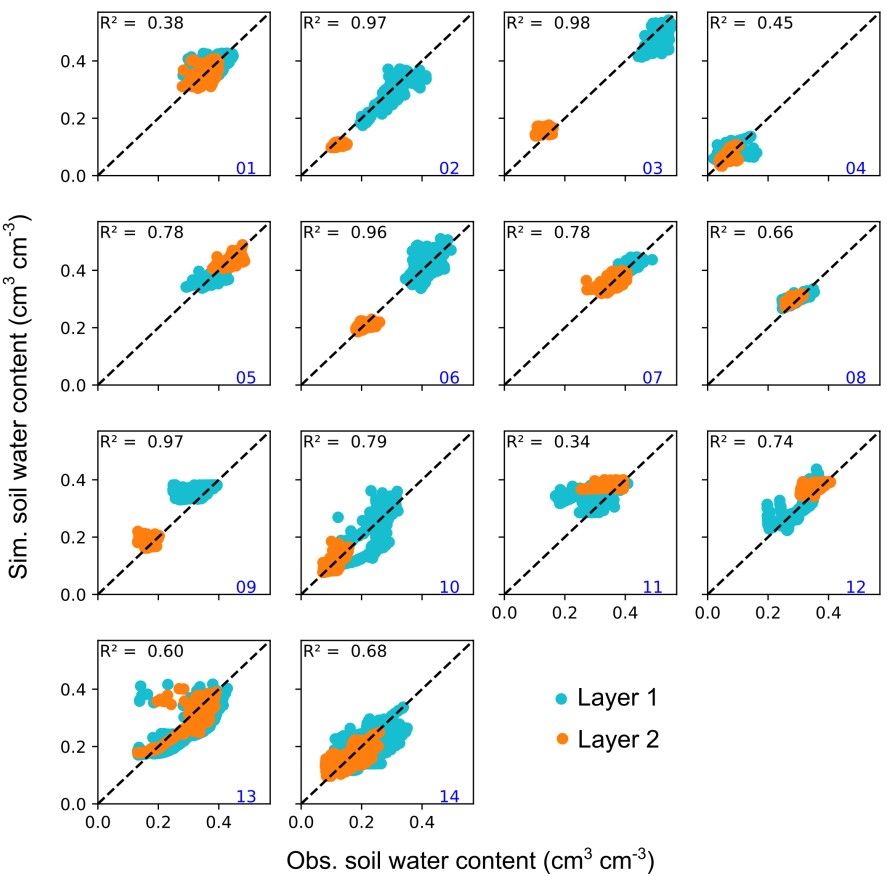

**Figure A4.** Model validation showing the coefficient of determination ($R^2$) and scatterplots of simulated and observed soil water content from upper and lower soil layer (layer 1 and 2, respectively) for the 14 sites: (01) Lauterach, (02) Leutasch, (03) Achenkirch, (04) Gschlössboden, (05) Sillianberger Alm, (06) Zettersfeld, (07) Elsbethen, (08) Gumpenstein, (09) Aichfeld-Murboden, (10) Kalsdorf, (11) Pettenbach, (12) Schalladorf, (13) Lobau, (14) Frauenkirchen. The dashed black Diagonal shows the 1:1 line. Validation periods are given in Table A2.

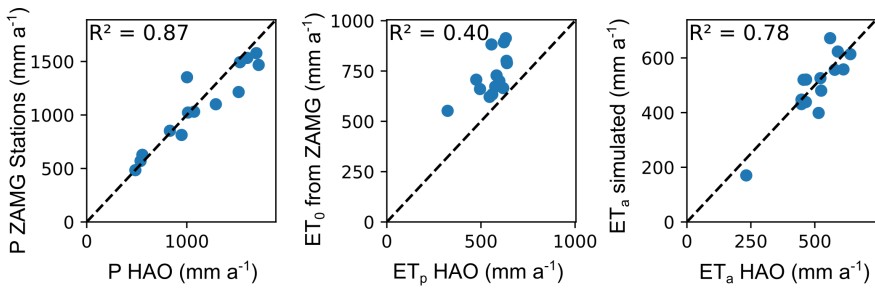

**Figure A5.** Scatterplots comparing the long-term averages of precipitation (P), potential and actual evapotranspiration ($ET_p$ and $ET_a$) from the digital Hydrological Atlas of Austria (HAO) (BMLFUW, 2007) with the corresponding rates of simulations in this study. The dashed black Diagonal shows the 1:1 line. Potential evapotranspiration in the HAO was calculated by Dobesch (2007) using the FAO approach described by Doorenbos and Pruitt (1977) resulting in lower values than those of this study which were calculated for a grass reference according to Allen et al. (1998).

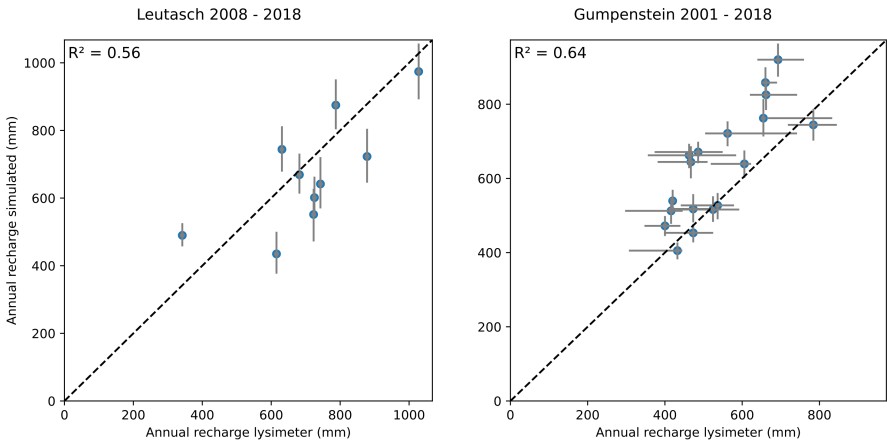

**Figure A6.** Model validation using lysimeter data from Leutasch and Gumpenstein. Scatterplots and coefficients of determination ($R^2$) are shown for simulated and observed annual seepage flow. Blue dots show median estimates and the grey errorbars depict the 95% credible interval from propagated parameter uncertainty. The dashed black Diagonal shows the 1:1 line. Leutasch seepage measurements are obtained from a single lysimeter; for Gumpenstein, the 95% uncertainty interval in lysimeter measurements was calculated from a cluster of four lysimeters.

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
