# Peer review of "Estimating vadose zone water fluxes from soil water monitoring data: a comprehensive field study in Austria"

_Hydrology and Earth System Sciences, 2022_

## Referee Comment (RC3)

[referee-annotated manuscript omitted]

---

## Author Response (AR1)

**Responses to Editor's and Reviewers' comments**

**Editor**

Dear Authors,

Your submission was evaluated by three reviewers who are well-focused on the topic but provided slightly different rates and comments mainly due to their different backgrounds. Overall, the scientific quality and significance of your study were evaluated quite well, whereas the presentation of the investigation requires some improvement. I support the general assessment of these reviewers. However, as also raised by one reviewer, I do suggest the uncertainty issue should be discussed more in-depth as it is an interesting question of your study that definitely can stimulate the interest of a wider readership toward your paper.

Your replies during the discussion phase showed that your group is acquainted with this question, but I think that treating epistemic uncertainty through an assessment of parameter sensitivity might resolve only an aspect of the problem. Maybe I miss something, but the question of parameter correlations and how it reflects on the propagation of uncertainty seems a bit overlooked. One question is to assess parameter uncertainty, whereas another question is to assess how epistemic uncertainty is reflected in the output of a model (a hydrologic one in this case). It is not always the case that uncertainty in the inputs is then lowered in the model outputs (as it can happen when comparing infiltration versus drainage processes). You may find it useful to discuss this latter question in the paper. Due attention should also be devoted to the comments from Reviewer #3.

I recommend reconsideration following major revisions, and invite you to submit your revised manuscript together with detailed point-by-point responses to all comments received thus far. Should you disagree with a comment or suggestion, please explain why clearly.

**Response:** We thank the Editor and all the Reviewers for their comments and suggestions, which have improved our manuscript. All comments have been thoroughly considered in our revisions. In particular, we included more details on the Bayesian uncertainty analysis by adding

- An improved description of the method
- An additional figure to discuss marginal posterior distributions and address the effect of parameter interdependencies/ correlations between parameter values in posterior samples on propagated predictive uncertainty
- Additional figures and discussion of propagated parameter uncertainties in cumulative and peak recharge estimation

As suggested by the editor, we address the topic of epistemic uncertainty in Lines 339-347 of the revised and marked manuscript:

**"Uncertainties in water fluxes were treated as aleatory, derived from stationary statistical characteristics. In addition, the epistemic uncertainty associated with the lack of knowledge about the correct representation of system dynamics (conceptual uncertainty) and forcing data may affect the overall predictive uncertainty and reduce the effective information content of observations (Beven, 2016). Due to their complex and often dynamic nature, epistemic uncertainties pose important conceptual and numerical challenges. For instance,**

**model conceptual uncertainty can be assessed by comparing different model structures using specific statistical metrics (e.g., marginal likelihood). This was, however, beyond the scope of the present study, which focuses on the inverse estimation of soil water fluxes at multiple monitoring stations to discuss implications on the water balance. In this framework, an appraisal of the model structural adequacy through posterior predictive checks appears sufficient."**

In the following, we describe our revisions note by note in response to the Reviewers' comments, indicating the line numbers of the marked manuscript.

We welcome the suggestion of the Editor to reconsider our thoroughly revised manuscript for publication.

**Reviewer #1**

I enjoyed this paper. The authors made use of an extensive set of water content profile time series in two differing climates within Austria. They used an accepted Bayesian inference algorithm to fit HYDRUS1D to the observations. They found that recharge is highly correlated with precipitation in the shorter term and aridity in the longer term. These results are well aligned with expectations from previous publications. My only complaint is that the method is based on a Bayesian approach, but there is almost no discussion of the resulting uncertainty of the inferences, how these depend on site conditions, how these might affect larger interpretations, and how the data did or did not constrain these uncertainties. The results and conclusions are primarily based on deterministic findings that seem to rely on the maximum likelihood parameter values. (Although this isn't discussed in detail.) I am actually fine with that - as stated above, I think that this makes a useful contribution. But, in the end, I was left wondering why the Bayesian approach was used rather than another parameter estimation tool. I would like to see the authors include some discussion of the special insights that resulted from the Bayesian analysis.

Best

Ty Ferre

**Response:** We thank Dr. Ferré very much for his appreciation of our work! We agree that a more detailed presentation and discussion of the results of the Bayesian analysis is helpful.

In our revised manuscript, we added a figure to show marginal posterior distributions of SHPs for one exemplary site. In this context we further discussed how parameter uncertainty was constrained by the observations in Lines 249-256:

**"A typical example for marginal posterior distributions resulting from SHP estimation on the basis of volumetric soil water content data in this study is shown in Figure 3 for the upper soil layer of the mountain location Zettersfeld. Limits of the plot axes are given by the prior bounds. This representation shows how well the calibration data constrained uncertainties of each parameter: the posterior range of $\theta_r$ is only slightly reduced compared to the prior range, indicating that $\theta_r$ was least sensitive for simulating soil water content and poorly informed by observations. The parameter $K_s$ has a wide posterior range (although clearly reduced compared to the prior), showing a logarithmic distribution and a clearly defined mode. On the other hand, the parameters $\alpha$, and especially $n$ and $\theta_s$, show narrow posterior distributions which appear leptokurtic, indicating a higher sensitivity for the soil water content simulations and a high information gain from the calibration data."**

We further included two figures to visualize the results of the uncertainty propagation for (1) cumulative recharge and (2) magnitudes of recharge peaks at all sites in the revised manuscript, and discussed them in Lines 318-335:

[revised manuscript text omitted]

The second part of the paper was aimed at characterizing the Austrian sites based on measured and modeled hydrological variables and site properties. In the revised manuscript, we clarified our approach in Lines 376-379:

**“In the following section we characterize the 14 monitoring according to hydrologically relevant properties including model estimations from the previous section. Since uncertainty in long-term actual evapotranspiration and recharge rates were generally low, and to enable the analysis with common statistical tools, we will proceed from here on using the median values without consideration of uncertainty ranges.”**

**Reviewer #2**

General remarks:

The manuscript is very well structured, very well written and represents an interesting scientific contribution to the determination of groundwater recharge rates. Changes to the manuscript are not necessary.

Specific comments:

The sites in Austria used for the study were described comprehensively, as were the statistical methods used. Why Bayes' theorem was used in the statistical analysis was sufficiently explained. In order to be able to better evaluate the results obtained, the methodological limitations were explained in detail.

Technical corrections: no corrections necessary

**Response:** We are very happy about the positive and motivating feedback on our work from Reviewer#2! We thank him/her for reviewing our manuscript.

**Reviewer #3**

Dear Editor, dear authors,

Please find below my review of the paper entitled "From soil water monitoring data to vadose zone water fluxes: a comprehensive example of reverse hydrology" by Marleen Schübl, Giuseppe Brunetti, Gabriele Fuchs, and Christine Stumpp.

This article investigates the use of the Bayesian approach to invert water content profiles to derive the soil hydraulic parameters, including their statistical distribution and related statistical parameters. Based on this information, the authors compute the water cycle over large periods and quantify the groundwater recharge, its uncertainty, and its temporal variability at 14 sites in Austria. The authors conclude that there is a West-East gradient with more continuous groundwater recharge at mountainous sites with wetter climates versus seasonal lower groundwater recharge in the Eastern plain.

The article is well-organized, well-illustrated, and well-written. The scientific question is properly stated, the methodology to answer conclusions is clear and straightforward, and the conclusions are quite obvious. The paper addresses an important topic and deserves to be published in the HESS journal. However, I have several concerns that should be addressed prior to publication.

**Response:** We thank Reviewer#3 for his/her appreciation of our paper and the helpful feedback.

I am not very familiar with the Bayesian approach, and thus hope that my comments do not reveal my lack of expertise in this subject. However, I consider that any paper should be standalone and present clear facts understandable by any scientific reader. Several points deserve to be clarified:

- For the Bayesian approach, the choice of distributions needs to be clarified. If the errors between the modeled and observed data are expected to obey the normal law, no details are given about the laws of the soil hydraulic parameters. I expect most parameters to follow normal laws and hydraulic conductivity to follow a log-normal law. If I understand well, the Bayesian approach allows us to characterize the SHP laws. Then, why not show them in the Result section and state on the alignment of normal laws? Why not state on the multimodality features of the SHPs? In addition, do the SHPS distributions have any consequences on the Bayesian approach and the Monte Carlo method? Is the normality of errors between experimental and modeled data compatible with any statistical law for SHPs?

  **Response:** The Reviewer raises an important point of discussion to better clarify our approach.

  The Bayesian inference can be applied directly to obtain Soil Hydraulic Parameters (SHPs), if data include pressure head, water content, and conductivity (e.g., laboratory measurements derived from the simplified evaporative method). Instead, our study uses field scale observations of volumetric water content to inversely estimate the most probable distribution of SHPs that generated observations. Therefore, the assumption of homoscedastic and normal errors (reflected in the likelihood function) refers to TDR measurements, not to SHPs. The posterior distribution of SHPs is inferred by combining two components: 1) the **prior** distribution, which reflects the modeler's believe about Soil Hydraulic Parameters (SHPs) **before** considering measurements (in our case, soil water content), 2) the likelihood, which describes the probability

that a parameter set drawn from the prior has generated the observations. By combining the likelihood and the prior with Nested sampling and HYDRUS (or Markov-chain Monte Carlo), we obtain a **posterior** distribution of the most probable SHP values, which reflects the parameters' uncertainty:

1. Prior: As it is frequently the case in vadose zone hydrology (e.g., Brunetti et al. 2020 https://doi.org/10.1016/j.jhydrol.2020.124681, Wöhling et al. 2015 https://doi.org/10.1002/2014WR016292), we assumed bounded uniform priors to avoid making important assumptions on the shape of the posterior, and let the data tell us what is its shape. However, at the same time, we imposed hard boundaries on the parameters to avoid the possibility to obtain physically unrealistic values.
2. Likelihood: We assumed that sensor errors are normally distributed. This is widely established approach in inverse vadose zone modeling (e.g., Schelle et al.. 2012 https://doi:10.2136/vzj2011.0169, Gao et al. 2019 https://doi.org/10.2136/vzj2019.03.0029).

- The problem of equifinality and non-uniqueness needs to be addressed in the paper. The authors inverted all the SHPs, except the parameter "l" fixed at 0.5. However, we know that many parameters may be interrelated, and some may have a poor impact on water fluxes. In particular, the value of the residual water content has no effect (or very little on the water fluxes), so this parameter may not be reachable when inverting. A similar statement may apply to the saturated water content. What is the strategy of the authors regarding this aspect of non-uniqueness? Why not propose a sensitivity analysis that shows the most influential parameters and select those when inverting water content data while suggesting additional information for the others?

**Response:** Also in this case, we thank the Reviewer for pointing this aspect, which gives us the opportunity to better clarify the utility of the Bayesian approach to address equifinality, and to discuss the limitations of the sensitivity analysis compared to the Bayesian probabilistic approach.

The Bayesian inference is precisely conceived to have a statistical rigorous appraisal of the "equifinality and non-uniqueness". The Bayesian approach infers a marginal posterior distribution that exposes the parameters' uncertainty, and their interaction (e.g., correlation). If the resulting uncertainty is high (i.e., wide posterior), then data are not informative for that parameter. In this case, the modeler has two choices: 1) ask for other measurements (e.g., disk infiltrometer for Ks) to have more informative priors and run again the Bayesian analysis; 2) honestly communicate what is the parameters' uncertainty with the data available, and more important, how the estimated uncertainty propagates in the model simulations. This is what we precisely did in our study.

The sensitivity analysis is certainly a valuable tool, which we use frequently in our research. However, it will not provide any more meaningful information compared to the Bayesian analysis for this work. A global sensitivity analysis will sample the parameters' space (frequently ineffectively as MCMC techniques are much better in finding high-probability regions), and then decompose the variance to identify influential and uninfluential factors. But this is already better targeted in a Bayesian analysis: influential parameters are those that exhibit leptokurtic

posteriors, while uninfluential factors are those that have similar prior-posterior distributions (in our case flat). The sensitivity analysis might have some utility in high dimensions for numerical sampling reasons, but this is not the case and is beyond the purpose of the present study.

In the introduction of the revised manuscript, we have added to the description of the Bayesian approach in Line 65:

**"Posterior parameter distributions also reflect the non-uniqueness and equifinality of parameter values."**

We further justified our choice of prior distributions and have added some clarification on the Bayesian method in Lines 185-191:

**"Prior knowledge, i.e. information available before looking at measured data, is included in the Bayesian inference via the prior distribution which can be chosen as a uniform density bounded by physical limits (e.g., Brunetti et al., 2020b; Gupta et al., 2022; Wöhling et al., 2015). In this study, uniform prior distributions were assumed for all parameters and sites. Their ranges were established based on texture information, literature review, and preliminary testing to prevent truncating posteriors. Final ranges are given in the Appendix in Table A3. By combining the likelihood and the prior, we obtain a posterior distribution of the most probable SHP values, which reflects the parameters' uncertainty."**

In our revised results section, we included an example of the marginal posterior distributions of SHPs and discussed them in Lines 249-263:

**"A typical example for marginal posterior distributions resulting from SHP estimation on the basis of volumetric soil water content data in this study is shown in Figure 3 for the upper soil layer of the mountain location Zettersfeld. Limits of the plot axes are given by the prior bounds. This representation shows how well the calibration data constrained uncertainties of each parameter: the posterior range of $\theta_r$ is only slightly reduced compared to the prior range, indicating that $\theta_r$ was least sensitive for simulating soil water content and poorly informed by observations. The parameter $K_s$ has a wide posterior range (although clearly reduced compared to the prior), showing a logarithmic distribution and a clearly defined mode. On the other hand, the parameters $\alpha$, and especially $n$ and $\theta_s$, show narrow posterior distributions which appear leptokurtic, indicating a higher sensitivity for the soil water content simulations and a high information gain from the calibration data."**

**Parameter interdependencies in the inverse estimation are reflected in the shapes of bivariate contour or scatter plots of posteriors (see Figure A2 in the Appendix for a representation of posteriors with closer axes ranges). By random sampling from the posterior, the effect of these correlations is propagated in the uncertainty in the prediction of soil water fluxes. Usually, a negative relation exists between the VGM shape parameters (e.g., Scharnagl et al. 2011; Romano and Santini, 1999; Vrugt et al., 2003). Here, both $\alpha$ and $n$ show narrow posteriors and stray very close to the lower physical bounds (0 and 1, respectively). The correlation of posterior samples for $\alpha$ and $K_s$ can be expected to have some effect on the uncertainty in recharge peak prediction, for which both parameters (but especially $K_s$ under wet conditions) are sensitive (Schübl et al., 2022)."**

- I also have some concerns regarding the data inverted to derive the SHPs. In their study, the authors invert only water content profiles. However, if I remember well, they also have water pressure head profiles for some sites. I understand they selected the water content profiles because they had those data at their disposal at all sites. However, for a given site (with the two types of data), they could have compared the results when inverting water content and water pressure head. My feeling is that the authors may not have had the same results. Based on this comparison, they might validate the choice of water content for all sites and strengthen their conclusions. That could be the topic of further research.

  **Response:** In general, we agree that it is helpful to include pressure head data, as it can help to identify SHPs with even less uncertainty than with soil water content data (Schübl et al., 2022). However, at the sites in this study we had some issues with soil pressure head measurements: (1) they were not available for all sites which would have impaired the comparability of results between locations (2) they were composite from different measurement techniques (tensiometers and gypsum blocks) and included sudden shifts and large gaps in time series. Altogether we found the measurements to not be reliable enough to be used in this study. We agree that results for SHP estimates might change when including the available soil pressure head data and have included this in the discussion (see reply to comments in manuscript). We also agree that this would be a very interesting topic for further studies with improved field measurements of soil pressure heads.

- Lastly, I had some questions and concerns about the ACP proposed at the end of the result section. I was surprised by the plots of "individuals" (i.e., sites) and the "variables" on the same plots (Figure 4). Even after searching on R tutorials and finding these types of plots, I am not convinced that we have the right to do so. For ACP, variables and individuals don't have the same nature and should be plotted on separate plots. I also suggest plotting the correlation circles and commenting only on the vectors (variables) that are well represented on the maps, i.e., which vector is close to the correlation circle.

  **Response:** We used this PCA Biplot here (Figure 7 of the revised manuscript) to visualize the two clusters of hydrologically similar sites in context with the variables according to which they have been characterized. This kind of visualization, with individuals (sites or samples) and variables in the same plot, has been used in several studies, for example by Luna et al., 2018 https://doi.org/https://doi.org/10.1002/eco.1896 (Figure 7), Rodríguez et al., 2020 https://doi.org/10.1007/s10750-020-04201-5 (Figure 3), Gibson et al., 2019 https://doi.org/https://doi.org/10.1016/j.ejrh.2019.100643 (Figure 7).

  Some R tutorials showing this kind of plot are

  1. https://f0nzie.github.io/machine_learning_compilation/detailed-study-of-principal-component-analysis.html (See 4.20 Biplot)
  2. https://www.datacamp.com/tutorial/pca-analysis-r
  3. https://finnstats.com/index.php/2021/05/07/pca/

  Citing from the first tutorial for the use of Biplots with variables and individuals, the focus is "…on the direction of variables but not on their absolute positions on the plot. Roughly speaking, a biplot can be interpreted as follows: an individual that is on the same side of a given variable has a high value for this variable; an individual that is on the opposite side of a given variable

has a low value for this variable." We wrote the code for our plot in Python using the sklearn module (Pedregosa et al., 2011 https://arxiv.org/abs/1201.0490) which is now cited properly in the revised manuscript.

We agree with the comment in the manuscript that the data is not well represented in the Biplot of Figure 4(b) and it does not add further insight, we therefore deleted it.

The authors will find an in-depth review in the enclosed file, with suggestions, comments, and proposals throughout the manuscript. Again, this paper is valuable and promising, and I have no doubts that it will be published after improvements.

**Response:** We thank the Reviewer for the positive assessment of our work. We are also grateful for the suggestions, which helped to improve the paper.

**Comments in Manuscript:**

L75: Why not also consider water pressure head profiles? Inverting other types of measures could change the features of estimates.

**Response:** In general, we agree that it is helpful to include pressure head data, as it can help to identify SHPs with even less uncertainty than with soil water content data. However, at the sites in this study we had some issues with soil pressure head measurements (see manuscript L105/6 and our reply comment). We agree that results for SHP estimates might change when including the available soil pressure head data, therefore we included this in the discussion in Lines 334-337:

**"[…] Especially the combination with soil matric potential measurements has been shown to be highly informative for SHP estimation and to considerably reduce uncertainties in recharge estimation (Schübl et al., 2022, Schelle et al. 2012). Including additional measurements in the analysis, however, might not only lead to different shapes in SHP posteriors, but to altogether different estimates. This issue requires further investigation with available soil water monitoring data."**

L103: Any rationale for the choice of the period?

**Response:** We in included the justification in Lines 106-110:

**"The length of calibration periods was chosen to be similar for all sites, long enough to be informative for a range of soil water conditions, and excluding the winter season which would require simulation of snow accumulation and melt processes. This exclusion allowed to reduce computational cost and numerical sensitivity of the simulations which often lead to non-convergence or delayed convergence of the sampling algorithm in the Bayesian analysis (described in Sect. 2.3)"**

L105/106 That is a good reason. We could have imagined using both water content and water pressure head profiles at some sites to compare the estimates and their distributions when one or the other signal is used. Which signal is the best? That may be a question for further studies.

**Response:** We investigated this question in Schübl et al. (2022) and came to the conclusions (based on synthetic data scenarios within the tensiometer range) that SHPs are identifiable and in general associated with less uncertainty when using matric potential as compared to soil water content measurements (see comment above). For the reasons highlighted, we did not use water pressure head data here, however, we agree that this would be a topic for further studies with improved field measurements.

**In Line 122 we have corrected the symbol.**

**In Line 134 we have removed the bracket.**

L144/145: You simplified the soil profile. However, weren't there some sites with more than three soil horizons? What could be the consequences of such simplifications on soil hydraulic characterization and water fluxes?

**Response:** We included in Lines 148-152:

**"The vast majority of the soil profiles indicated a distinct topsoil overlying deeper soil layers that had low to mild degrees of inhomogeneity. Dealing with 14 monitoring stations, we uniformly adopted two soil layers with varying thickness across different locations, aiming to reduce the overall computational burden of the Bayesian analysis while maintaining a physically realistic description of the soil domain. Simplifications of the soil profile in the model geometry with a mildly heterogeneous soil will usually lead to an acceptably small loss of accuracy in effective parameters (Schneider et al., 2013)."**

L147: Could you cite works that justify that the zero flux plane (ZFP) must be above 1.5m depth in all your sites? It seems reasonable to think that 1.5m is enough to include the ZFP and estimate GW recharge. But it is better to cite at least one source.

**Response:** In Lines 156-158 we included:

**"Similar to our approach, Simunek (2015) and Heppner et al. (2007) simulated groundwater recharge with HYDRUS-1D for grass-covered soils as bottom flux at 100 cm profile depth; Assefa and Woodbury (2013) used different profile depths of up to 150 cm."**

L158/159: Unclear. Does that concern only the simulation of the six-month period? If yes, please clarify your choice in terms of simulated chronics here.

**Response:** We rephrased this in Lines 167-169**:**

**"For inverse parameter estimation during the half-year calibration periods, as well as for the model validation periods, we chose boundary conditions with respect to the conditions at the measurement plots, i.e. seepage face for the lysimeter sites and free drainage for sites with natural field conditions."**

L166/167: Do you have any reference to justify such linearity between -2 and +2°C? I am not convinced by such a hypothesis given the complexity of processes (freezing rain, isothermal snowfall, etc.).

**Response:** We included a justification in Lines 178-180:

**"This default snow routine in HYDRUS is based on assumptions by Jarvis (1994) and has been found to be suitable for estimating soil water fluxes in unfrozen soils in several studies (e.g., Assefa and Woodbury, 2013; Zhao et al., 2008)."**

L169: I am fine with this part, but we should somehow address the problem of parameter equifinality and non-uniqueness of solutions. May we, by increasing a parameter, compensate for the effect of the other? This point is somehow addressed when investigating the correlation matrix between parameters in the result section but not evoked in the modeling section. I suggest giving more details in this section about the distributions of each hydraulic parameter and the random sampling for the Monte Carlo method.

**Response:** We thank the Reviewer for his/her comments on this section. In the revised manuscript, we clarified the Bayesian approach, justified our choice of prior distributions and discussed resulting marginal posterior distributions (see reply to main Review comment).

L172: Couldn't we note this as P(D | \Omega, M), given that the model is chosen as the first step, and then the data D and the parameter \Omega are related against each other (for a fixed model)? I would prefer the notation P(D| \Omega, M) and P(M | D, M) all together.

**Response:** We used the common notation for the purpose of Bayesian parameter estimation with a previously chosen model (see e.g., Skilling, 2006 https://doi.org/10.1214/06-BA127).

L182: Do we speak of the distribution of randomly sampled values for each parameter or the distribution of hydraulic parameters on the field? Should we consider uniform distributions for all hydraulic parameters? Alternatively, shouldn't we consider a log-normal distribution for Ks and normal distributions for the other parameters? Which impacts does the choice of the distribution have on the likelihood function and on the Monte Carlo method?

**Response:** The prior distribution reflects the modeler's prior belief about the parameter before running the analysis, and should be supported by available information. The normal distribution implies that we have data-supported guesses about the mean and the standard deviation of the parameter. However, we didn't have these supporting information, therefore we assumed flat priors to let data inform us about the parameters' values.

Impact of the prior on the Monte Carlo procedure: if the prior is close to the posterior, the convergence will be faster. Otherwise, samples drawn from the prior will be frequently rejected as they don't belong to the posterior, and the convergence will be delayed.

The choice of the prior has no impact on the likelihood.

L193: "multimodal distributions" related to the SHPs?

**Response:** It relates to the shape of the posterior, which can have (possibly) multiple modes (i.e., high-likelihood regions). These distributions are problematic to sample but MULTINEST does that fairly good.

L215:  Why not 0.05, as usual?

**Response:** E.g., Moeck et al., 2020 (https://doi.org/10.1016/j.scitotenv.2020.137042) use a 90% confidence level for similar questions (even with a much greater data set).

Table 1 caption: Please remind us why the credible intervals are not symmetrical while parameter distributions may be quasi-gaussian (except for Ks).

**Response:** We do not expect that posterior probability distributions of SHPs are necessarily quasi-gaussian. Therefore, uncertainty bounds below and above the median are given separately (instead of being summarized as standard deviation). We included an example of marginal parameter posterior distributions to better illustrate this (see above comments).

Figure 3: Why not distinguish the dots for the two layers?  Why don't you show the trends for these parameters? It could be interesting to discuss the differences between SHPs.

**Response:** We distinguished the two layers and included residual and saturated water content parameters in the plot (Figure 4 of the revised manuscript). We added in the discussion in Lines 271-279 (see reply to comment of Reviewer#1)

L291: "agreement" instead of "fit" which is more dedicated to the alignment of a given model to experimental data.

In Line 367, we replaced fit by agreement.

L294-295: We have one order of magnitude of difference. Does that stem from the representativeness of the lysimeter, as said below?

**Response:** We suggested that potential reasons for the larger variability in the lysimeter measurements are spatial heterogeneities and uncertainty in seepage measurements (as stated below).

Figure 4: For PCA analysis, theoretically, it is not correct to superpose the plots of individuals and the correlation circle. You can do it when individuals and variables are of the same nature, which is the case of MCA (Multiple Correspondance Analysis) but definitely not the case of PCA. Then, I reckon that you explain how you did your graphs, or alternatively, I reckon that you replace your figure 4 with a new plot with separated subplots of the plans (F1, F2), (F3, F4) and the correlation circle (F1, F2), (F3, F4). Ensure to include the unity circle that indicates if the variable is well represented in the plan or poorly represented. For instance, GWR/P, GWR, and P are poorly represented in the plan (F3, F4).

**Response:** We used this PCA Biplot here to visualize the two clusters of hydrologically similar sites in context with the variables according to which they have been characterized (not for further statistical analysis). This kind of visualization with individuals (sites or samples) and variables in

the same plot has been used in several studies (see discussion comment for references and links to R tutorials). We wrote the code for our plot in Python using the sklearn module which is now cited properly in the Figure description. It was based on these tutorials:

https://ostwalprasad.github.io/machine-learning/PCA-using-python.html

https://blog.bioturing.com/2018/06/18/how-to-read-pca-biplots-and-scree-plots/

We agree, however, that the Figure 4(b) for PC3 and PC4 does not represent the data very well and does not give much further insight, we therefore deleted it.

Theoretically, in PCA analysis, all the variables are independent. Then, we avoid designing new variables from already considered variables because it reduces the degree of freedom of the system and defines an ill-posed problem (as for inversion). Please, address this point in the revised version of the manuscript.

**Response:** To our knowledge, there is no requirement that variables included in a PCA must be independent. We used this representation for displaying the site clusters and to describe their characteristics, not for further statistical analysis. We believe this representation is applicable for this purpose.

L327/28: Caution: you have more mountainous sites in the western part of the studied area. Thus altitude and longitude are correlated, which biases the analysis.

**Response:** We rephrased this statement in Lines 407-410:

**"Precipitation and recharge rates were higher in the West than in the East, following both the longitudinal gradient in altitude and the climatic influence of the wet oceanic climate in the West, with high precipitation and recharge rates even at lower altitudes (Lauterach, Elsbethen), versus the dry continental climate in the East."**

L333-335: Awkward. Correlation does not give information about the function that relates the variables. Please, elaborate.

**Response:** We deleted this half-sentence in Line 414:

**"The fraction of potential groundwater recharge to precipitation (GWR/P) was strongly correlated with the amount of precipitation (r = 0.91). Similarly, Barron et al. (2012) found […]"**

L363: See below my inputs in the annexes. Some events were missed (even if roughly the modeling did well).

**Response:** We included the discussion on missed events by the model in Lines 295-297:

**"Some events were missed by the model: at Lauterach and Elsbethen, the drying of the lower soil layer in summer was underestimated; at Gschlössboden, the peak in soil water content in the early calibration period was missed for both layers."**

L380-382: Please, add, afterward, a few words on perspectives of improvement, like the combination with data at lower depths (GW monitoring) and the consideration of additional physical processes (preferential flows and water repellence, etc.).

**Response:** We added perspectives of improvement in Lines 461-465:

**"The approach could be improved by including information on the deeper vadose zone to obtain more insight on temporal variation and seasonality of actual recharge, and to improve the model structure including lower boundary conditions. Especially at dry locations, using improved and additional measurements (e.g. of soil matric potential) could help reduce uncertainty in cumulative recharge estimation. Additionally, consideration of sites with varying slopes and the inclusion of surface runoff simulations in the analysis might improve representativeness for larger scale."**

From the methods and results in this study, we cannot draw specific conclusions on which physical processes must be considered in the modeling, as this would require separate case studies for each individual monitoring site and is beyond the scope of this work.

In the Appendix, we improved the Figure descriptions including the 1:1 lines.

---

## Author Response (AR2)

**Responses to Editor's and Reviewers' comments**

**Editor**

Dear Authors,

Your submission is close to the final acceptance. The revised paper has received additional, but useful (I guess), comments from reviewers. Please, take a look at these comments and see if they can still be included so as to improve or make clearer a few parts of your study.

Send me your review and a document explaining the changes done. Should you disagree with some comments, or think a change is unfeasible, please explain why.

I look forward to receiving your documents.

**Response:** We thank the Editor for appreciating our revisions and the suggestion to accept the paper after minor revisions. We thank all the Reviewers once more for their comments and suggestions, which further improved our manuscript.

Our revisions in response to Reviewers' comments are described and justified below. We indicate the line numbers of the marked manuscript.

**Ty Ferre (Reviewer #1)**

I want to start by saying that I really like what the authors have done. They have taken an unusually comprehensive and well-characterized data set and subjected it to improved inverse analysis. I have no objection to this paper being published as is ... except that it was not at all what I was expecting based on the title.

The fundamental problem here is that there is no good way to measure recharge flux. As a result, all that the authors can do is to look at the statistics of their predictions. How can a reader trust that the estimated fluxes are accurate based on the finding that your inverse method gave small uncertainties (in some cases) compared to the mean predicted values? Again, this is the way that it is right now ... we don't have a valid ground truth. But I would have been much more comfortable with the paper if the authors had presented it as a field study of recharge and then commented on the strengths and weaknesses of their selected analyses.

Not to overstate it, but what is the purpose of highlighting 'reverse hydrology' in the title. The word 'reverse' only shows up one more time in the paper and in an entirely different context! If the paper were refocused very slightly to better represent what (I think) it is, I would suggest accepting as is!

Best

Ty Ferre

**Response:** We thank Ty Ferre for appreciating our revisions and the helpful feedback concerning the title! We have adapted it to "Estimating vadose zone water fluxes from soil water monitoring data: a comprehensive field study in Austria". This leaves out the term "reverse" and makes the scope of the study clearer.

**Jasper Vrugt (Reviewer #4):**

Review of "From soil water monitoring data to vadose zone water fluxes: a comprehensive example of reverse hydrology"

I have been asked by the Editor to provide a re-review of this paper. I looked at the first round of comments of the other reviewers and the document with track changes. The paper is generally well written and addresses an important topic in hydrology, namely the quantification of groundwater recharge rates and their associated uncertainty. The paper makes a useful contribution. I recommend a major revision. I list my comments - not in any particular order.

**Response: We thank Jasper Vrugt for reviewing our manuscript and sharing his expertise on Bayesian inference. We have considered all comments which have been very helpful for improving the paper. In the following, we address our changes based on the comments and/or justify our choices.**

0. Reverse hydrology? We have hydrology backward; inverse. Reverse hydrology is catchy but I personally would stick to jargon of inverse. Also, I am not so sure that the example is very comprehensive; comprehensive in analyzing different sites, but not comprehensive in numerical modeling, inverse estimation, and uncertainty quantification. I'll discuss this further below.

**Response: We have changed the title to "Estimating vadose zone water fluxes from soil water monitoring data: a comprehensive field study in Austria" to refer the term "comprehensive" to the multiple sites representing different hydrological conditions and to leave out the term "reverse hydrology".**

1. Line 125: Units of S are missing

**Response: The units have been included (Line 126).**

2. Line 148: "The vast majority of the soil profiles indicated a distinct topsoil overlying deeper soil layers that had low to mild degrees of inhomogeneity" How did you determine this? Soils that may appear homogeneous visually, can be highly heterogeneous.

**Response: We determined this from the available soil water measurements and profile information (texture data and soil horizons) established by/for the Austrian ministry who operate the soil water monitoring network. We did not rely on a visual assessment. In Lines 149-151 we clarified this: "The available soil water measurements and profile information (texture data and soil horizons) indicated a distinct topsoil overlying deeper soil layers with low to mild degrees of inhomogeneity at the vast majority of the soil profiles."**

3. Line 167-169: "For inverse parameter estimation during the half-year calibration periods, as well as for the model validation periods, we chose boundary conditions with respect to the conditions at the measurement plots, i.e. seepage face for the lysimeter sites and free drainage for sites with natural field conditions."

First of all, you can remove the word "inverse" in front of parameter estimation. Secondly, I would argue that a six-month calibration period may be too short to get a) an accurate characterization of the soil hydraulic properties - let alone their uncertainty, and b) to remove the dependence of the

initial soil moisture state (=wetness of profile) and the resulting parameter estimates. This is a serious issue and authors need to demonstrate that their parameter estimates are not too dependent on the initial wetness; otherwise the inference and uncertainty estimates depend on the choice of the initial state. Not desirable.

**Response:** We removed "inverse" from the sentence (Line 169).

- We restricted the calibration to half-year since most of the sites are influenced by snow during winter. Snow simulation and parameterization of the snow routine introduce additional numerical burdens with more frequent non-converging model runs as well as additional complexity and potential biases in the calibration. Also, the use of spring-summer months, which have an alternation of wet-dry periods, is expected to increase the informativeness of soil water measurements.

- We used a model spin-up period of two months to relax the effect of initial conditions on the estimation procedure. We apologize for not including this detail in earlier versions of the manuscript. We have now mentioned it in the text and made the justification of our choice of calibration periods clearer, see Lines 105-111: **"The length of calibration periods was chosen to be similar for all sites, long enough to be informative for a range of soil water conditions. We excluded the winter season requiring the simulation of snow accumulation and melt processes as it increases the computational cost and numerical sensitivity of the simulations and introduces additional complexity and potential biases in the calibration. The use of spring-summer months, which have an alternation of wet-dry periods, is expected to increase the informativeness of soil water measurements."**

4. Line 182: Bayes theorem ... (remove "The")

**Response:** We removed "The" before "Bayes theorem" in Lines 70 and 184.

5. Why does Equation 7 not appear after Line 182-183? Unusual

**Response:** We shifted the equation to the indicated lines.

6. Some call $P(D \mid M,\Omega)$ the data likelihood but this is really the conditional probability as the parameters are assumed given (appear on right hand side of "|"), likelihood you write as $L(\Omega|D,M)$.

**Response:** We changed the description in Line 189: **"… $P(D \mid M,\Omega)$ is the conditional probability of the data given the model and parameters…"** and changed the notation of the Likelihood to $L(\Omega|D,M)$.

7. Measurements errors are assumed to be IID and lead to the standard normal likelihood. a) These assumptions are questionable at best and should be verified a-posteriori using diagnostic checks of the residuals (histogram of residuals, ACF and QQ plots); b) the posterior parameter distribution is strongly dependent on the choice of likelihood function - and, thus, the uncertainty estimates of the recharge rates are suspect. I would strongly recommend using a distribution-free likelihood function instead. This will adapt to the residual properties at hand; hence, satisfy residual

assumptions                    made.                    For                    example,                    check:
https://www.sciencedirect.com/science/article/pii/S002216942201112X

The universal and generalized likelihood functions are your best bet to getting the most accurate estimates of recharge uncertainty.

**Response:** We thank the Reviewer for suggesting these new likelihood functions. Here are the main justifications for our choice:

- The calibration procedure includes only volumetric water content measurements from TDR sensors. While the deterministic part of the measurement signal is correlated, the stochastic part (i.e., measurement error) is not as it is based on an electromagnetic instantaneous pulse. The correlated part of the signal is implicitly described when numerically solving the Richards equation, as the variables (theta and h) at time step t+dt are solved using their counterparts at time t. Therefore, we decided to use a more physically realistic likelihood to carry out a process-based probabilistic inference. Significant discrepancies between model predictions and observations are used as indicators that the model structure needs to be improved. On one hand, we agree that this approach might be restrictive, however, on the other, we think it might better target model inadequacies, which can be masked when using other likelihoods. Nevertheless, we thank the Reviewer for suggesting the universal likelihood. This is certainly something we want to explore in future studies!

- We have added a plot showing the diagnostic checks of the model residuals as example for Gumpenstein (layer 1 upper graphs; layer 2 lower graphs) where we do not see severe violations of our assumptions.

In Lines 197-199 we added: **"We used volumetric water content measurements from TDR sensors in the calibration where the measurement error is based on electromagnetic instantaneous pulses and can be assumed to be independent, homoscedastic, and normally distributed. This leads to a Gaussian likelihood function […]"**.

In Lines 204-210 we added: **"The choice of likelihood function is critical to the outcome of Bayesian inference and is the subject of ongoing debate. A recent promising approach that should be explored in future studies is the universal likelihood proposed by Vrugt et al. (2022). Instead of making prior assumptions about the distribution of model residuals in the likelihood function, this approach is distribution-adaptive to the actual residual properties. However, in the present study, we used the Gaussian likelihood function as described above for process-based probabilistic inference, where we use significant, systematic discrepancies between model predictions and observations that violate our assumptions as indicators that the model structure needs improvement. We show the residual checks as example for the location Gumpenstein in the Appendix (Fig. A1)."**

[Figure]

8. Subscripts that are acronyms should not be italic. "s" in theta_s, should be upright, otherwise it is considered a variable. This comment applies to all super/subscripts in the paper.

**Response:** We changed the subscripts in question to non-italic.

9. The authors use the MULTINEST sampling algorithm - I do not want to be difficult, but I would recommend the authors to have at least a quick look at the DREAM algorithm. This has been developed within the context of hydrologic problems - and is much better benchmarked than the MULTINEST algorithm. In fact, I am not sure if this algorithm has ever been used for hydrologic problems ; if so then it is important to show that it actually infers the correct parameter distributions - it should, but this is not a guarantee. Note that DREAM can also handle multimodal surfaces. This is demonstrated in theory and practice in related manuals and papers. The DREAM algorithm also provides estimates of the evidence - see Volpi et al. https://agupubs.onlinelibrary.wiley.com/doi/full/10.1002/2016WR020167

**Response:**

- The MULTINEST algorithm has been tested and benchmarked with hydrological models in previous studies (https://doi.org/10.1016/j.watres.2020.115973, https://doi.org/10.1016/j.jhydrol.2020.124681), (https://doi.org/10.1016/j.jhydrol.2018.06.055). In Schübl et al. 2022 (https://doi.org/10.1016/j.jhydrol.2022.128429) we tested MULTINEST with artificially generated data and similar HYDRUS models, where the algorithm reliably inferred the true parameter values as well as standard deviations of the artificial errors in the calibration data. This is mentioned in Lines 216-218 (**"The Nested Sampling algorithm as proposed by Skilling (2006) has been used successfully for parameter estimation and uncertainty quantification in studies with non-linear hydrological or biogeochemical models (Brunetti et al., 2020a; Elsheikh et al., 2013).")** We further added in Lines 218-220: **"It has been tested in Schübl et al. (2022) with synthetic data scenarios for SHP estimation**

**with similar HYDRUS models where it reliably inferred the true parameter values as well as standard deviations of the artificial errors in the calibration data.”**

- Allison et al. (2014) (https://doi.org/10.1093/mnras/stt2190) compared Nested Sampling, classic MCMC Metropolis-Hastings, and the affine invariant MCMC ensemble sampler, which shares multiple similarities with DREAM and DE-MC. They found that Nested Sampling delivers high-fidelity estimates for posterior statistics at low computational cost, and has comparable accuracy to MCMC techniques.

Therefore, based on our experience and existing studies, we trust that MULTINEST can provide reliable estimates of the posterior distribution.

10. Per my earlier point; The uncertainty estimates of Table 1 are strongly dependent on the likelihood function and, possibly, the choice of initial conditions. I believe this paper would be substantially stronger if this were investigated in more detail - and the likelihood function traded for a distribution-free formulation.

**Response:** Please refer to the previous responses.

11. Fig. 2b: I am a bit surprised that the posterior distribution of theta_r is relatively well defined. The soil moisture observations in the left plot show that the profile is quite wet during the calibration period; soil moisture values do not go lower than about 0.22. This is much larger than the residual moisture content, meaning that there will be hardly any information in the soil moisture observations for estimating theta_r. Hence, I would expect a much larger posterior uncertainty of this parameter - extending over almost its entire prior parameter range; unless the range of alpha and n are chosen so that high values of theta_r are discouraged. Certainly, theta_r plays a role in characterizing soil moisture flow at low moisture contents.

**Response:** In the old version of the manuscript, Figure 2(b) was a summary of the resulting relative uncertainty ranges for the parameters at all 14 sites while 2(a) on the left showed an example for the calibration only for the site Gumpenstein. To avoid confusion, we separated these two figures in the newly revised manuscript (Fig. 2 and Fig. 3). We agree that given the wet calibration period at Gumpenstein we expect theta_r to show a large posterior uncertainty. As given in Table 1, the uncertainty ranges for theta_r in Gumpenstein (according to the 95% confidence intervals) were 1.3 – 7.8% in the top soil layer and 1.7 – 10.1% in the bottom soil layer. The uncertainty ranges were thus quite large. The respective prior ranges for alpha and n are given in the manuscript (0.0001-0.5 and 1.01-2.70, respectively); they were not chosen to discourage high values of theta_r.

12. On a related note, my personal experience suggests that MCMC-HYDRUS sampling is difficult due to the numerical errors of HYDRUS - this results in very low acceptance rates; requiring an efficient MCMC method to traverse the many pits in the response surface introduced by the numerical errors of HYDRUS. What is the acceptance rate of MULTINEST? How many posterior samples do you have? And how do we know that the algorithm has formally converged? The advantages of multi-chain methods such as DREAM is that you can much better assess convergence of the chains by looking at the within and between-variance of the parameters (univariate scale reduction factor). The multivariate scale-reduction factor compares the covariances as well.

**Response:**

- HYDRUS errors: Numerical errors from HYDRUS that can lead to difficult posterior sampling are: 1) numerical diffusion, 2) non-convergence due to improper settings, 3) mass balance. The numerical diffusion was limited by adopting a relatively fine mesh, refined at the top to accommodate pressure gradients induces by atmospheric conditions. A crucial value to reduce the number of non-convergent HYDRUS runs is hCritA. If set too high, it will lead to floating precision error in the solver and non-convergence. We implemented a subroutine that set this value based on the soil hydraulic parameters proposed by the Bayesian sampler. In particular, hCritA is set equal a pressure that leads to a volumetric water content slightly higher than the residual water content. This drastically reduced the number of non-convergent runs. Finally, a large negative log-likelihood value was attributed to simulations affected by high mass balance error (>5%).

- Nested Sampling uses a different sampling approach than a classical MCMC scheme, therefore the acceptance rate does not have the same meaning here (it is supposed to decline with each iteration as the sampling approximates the bulk of the posterior). With Nested Sampling, the convergence is monitored via the accumulation of the evidence integral and the remaining prior volume. We describe this in lines 232-236 of the manuscript: **"At each iteration of the algorithm, the current maximum likelihood sample point is multiplied with the remaining prior volume to estimate the maximum remaining volume of the BME integral. Sampling is then terminated according to a tolerance (convergence) criterion, which defines when the remaining contribution from the current live points to the integral is considered to be small enough. At this point, it is expected, that the bulk of the posterior has been sampled sufficiently. The tolerance parameter in this study was set to 0.5"**. More detailed information is given in the papers by Feroz et al. (https://doi.org/10.1111/j.1365-2966.2007.12353.x, https://doi.org/10.1111/j.1365-2966.2009.14548.x).

- The number of posterior samples depends on the algorithm convergence, which was different for each monitoring station. On average, 4100 samples were used to characterize the posterior, which was randomly sampled 100 times to propagate the posterior uncertainty in model simulations. We have added further details, see Lines 237-240: **"The number of posterior samples provided by MULTINEST depends on the algorithm convergence with each model. On average, we obtained 4100 posterior samples and corresponding sample weights to characterize posterior parameter distributions. We used 100 random samples from the posterior to propagate parameter uncertainty in the model for long-term simulations to quantify the resulting uncertainty in recharge simulations."**

13. Line 306 - 309: "Overall, the validation of the models was acceptable with RMSE values ranging between 0.014-0.067 cm3 cm−3. Scatterplots including the coefficients of determination R2 (0.34– 0.98) for the validation period are shown in Fig. A3 in the Appendix." How did you determine that validation behavior was acceptable? I find the RMSE of 0.067 quite large; much larger than the measurement error of the data. As the authors discuss, this is a result in part of

measurement errors of rainfall / boundary conditions; This reiterates the importance to evaluate the likelihood assumptions using diagnostic tests of the residuals.

**Response:** We removed the word "acceptable" from this sentence and rephrased it to **"Overall, in the validation periods RMSE values ranged between…"** (Lines 323-324). We agree that 0.067 cm³/cm$^{-3}$ is a quite large error in the validation. It was found for the lysimeter site Pettenbach. We discussed the reasons for this specific case in the manuscript (Lines 318-322): **"At the Pettenbach lysimeter station, a crop rotation including fertilization was applied. It is possible, that this affected soil properties, which were assumed to be constant in the modeling. For example, Lu et al. (2020) showed in their review that root growth and decay can alter soil hydraulic properties; Whalley et al. (2005) found, that growing different plants had a significant effect on the porosity of the soil aggregates, and Schjønning et al. (2002) observed the development different pore systems in soils depending on crop rotation and fertilization."**

14. Figure 5 and 6 document only the impact of parameter uncertainty on the bottom boundary flux. But what about model uncertainty? This will make the credible intervals much larger. I think it is worthwhile to consider model uncertainty as well.

**Response:** We agree that it would be very interesting to comprehensively assess parameter and model structural uncertainty, e.g. in the framework of a Bayesian Model Averaging analysis (BMA) with multiple soil hydrological models/model structures. It is true that in this work we address only the propagated parameter uncertainty originating from the inverse estimation with one model. For an analysis including multiple models/ model structures we would have to focus on one or few study sites. Our focus here was to evaluate and compare soil water fluxes and parameter uncertainties from 14 different sites with the same estimation technique. We discussed this and other limitations of our study in the manuscript in Lines 355-372.

15. Section 3.3 is a nice part of this paper - trying to relate what has been found to soil properties, etc.

**Response:** We thank the Reviewer for appreciating this part of the manuscript!

I hope these comments are useful to further improve this paper,

Jasper Vrugt

Irvine, Feb. 15, 2023

**Response:** We thank Jasper Vrugt again for appreciating our work and the useful comments which helped improve our paper!

---

## Author Response (AR3)

**Response to Editor**

**Editor**

Dear Authors,

Taking into account your thoughtful replies to the reviewers' comments and further changes made to the text, your latest version of the manuscript can now be accepted as-is.

> **Response:** We would like to thank the Editor for his guidance throughout the revision process, for appreciating our revisions and comments and for accepting the latest version of our manuscript for publication!